# Genetic characterization of a unique neuroendocrine transdifferentiation prostate circulating tumor cell-derived eXplant model

Vincent Faugeroux[1,2], Emma Pailler [1,2,10], Marianne Oulhen[2,10], Olivier Deas[3,10], Laura Brulle-Soumare [3], Céline Hervieu[1,2], Virginie Marty[4], Kamelia Alexandrova[5], Kiki C. Andree [6], Nikolas H. Stoecklein [7], Dominique Tramalloni[5], Stefano Cairo [3], Maud NgoCamus[8], Claudio Nicotra[8], Leon W. M. M. Terstappen [6], Nicolo Manaresi [9], Valérie Lapierre[5], Karim Fizazi[1,8], Jean-Yves Scoazec [4], Yohann Loriot[8✉], Jean-Gabriel Judde[3] & Françoise Farace [1,2✉]

Transformation of castration-resistant prostate cancer (CRPC) into an aggressive neuroendocrine disease (CRPC-NE) represents a major clinical challenge and experimental models are lacking. A CTC-derived eXplant (CDX) and a CDX-derived cell line are established using circulating tumor cells (CTCs) obtained by diagnostic leukapheresis from a CRPC patient resistant to enzalutamide. The CDX and the derived-cell line conserve 16% of primary tumor (PT) and 56% of CTC mutations, as well as 83% of PT copy-number aberrations including clonal *TMPRSS2-ERG* fusion and *NKX3.1* loss. Both harbor an androgen receptor-null neuroendocrine phenotype, *TP53, PTEN* and *RB1* loss. While *PTEN* and *RB1* loss are acquired in CTCs, evolutionary analysis suggest that a PT subclone harboring *TP53* loss is the driver of the metastatic event leading to the CDX. This CDX model provides insights on the sequential acquisition of key drivers of neuroendocrine transdifferentiation and offers a unique tool for effective drug screening in CRPC-NE management.

[1] INSERM, U981 "Identification of Molecular Predictors and new Targets for Cancer Treatment", 94805 Villejuif, France. [2] Gustave Roussy, Université Paris-Saclay, "Circulating Tumor Cells" Translational Platform, CNRS UMS3655—INSERM US23 AMMICA, 94805 Villejuif, France. [3] XenTech, 91000 Evry, France. [4] Gustave Roussy, Université Paris-Saclay, Experimental and Translational Pathology Platform, CNRS UMS3655–INSERM US23 AMMICA, 94805 Villejuif, France. [5] Gustave Roussy, Université Paris-Saclay, Department of Cell Therapy, 94805 Villejuif, France. [6] Medical Cell Biophysics Group, Technical Medical Centre, Faculty of Science and Technology, University of Twente, 7522 NB Enschede, The Netherlands. [7] Department of General, Visceral and Pediatric Surgery, Medical Faculty, University Hospital of the Heinrich-Heine-University Düsseldorf, Düsseldorf, Germany. [8] Gustave Roussy, Université Paris-Saclay, Department of Cancer Medicine, 94805 Villejuif, France. [9] Menarini Silicon Biosystems S.p.A, 40013 Bologna, Italy. [10] These authors contributed equally: Emma Pailler, Marianne Oulhen, Olivier Deas. ✉email: Yohann.loriot@gustaveroussy.fr; francoise.farace@gustaveroussy.fr

Prostate cancer is the most frequent malignancy and the second cause of cancer-related deaths in men in Western countries. The vast majority of primary prostate cancer has a luminal adenocarcinoma phenotype and requires hormonal exposure to gonadal androgen for cell growth. The standard-of-care treatment for prostate cancer is androgen deprivation therapy (ADT) which results in inhibition of androgen receptor (AR) signaling pathway and efficiently controls the growth of androgen-dependent tumors. However, conventional primary ADT is only transiently effective, and most prostate cancers universally progress after a variable period of time to a status known as castration-resistant prostate cancer (CRPC) which is currently incurable. The treatment of CRPC has evolved in recent years with the advent of new active drugs with proven survival benefit including the chemotherapy agent cabazitaxel[1], two AR pathway inhibitors, abiraterone acetate[2] and enzalutamide[3], the immunotherapy sipuleucel-T[4] and the radiopharmaceutical radium-223 chloride (Ra-223)[5], along with taxane-based chemotherapy[6]. However, almost all patients develop resistance to these agents. The transformation of advanced CRPC into an aggressive neuroendocrine phenotype (CRPC-NE) with low or null AR expression is increasingly recognized as a mechanism of resistance observed in a subset of patients treated with AR-directed therapies[7–9]. In these patients, the development of effective therapeutic strategies has been hindered by an incomplete understanding of the mechanisms of transformation into CRPC-NE and the lack of experimental models.

The low number of CRPC experimental models constitutes a limitation to comprehensive understanding of CRPC biology and drug resistance. Despite its prevalence, prostate cancer has proven difficult to propagate in vivo and in vitro. Only seven prostate cancer cell lines are available, and they do not reflect the biological diversity of CRPC. Using a 3D organoid system, long-term cultures amenable to genetic and pharmacologic studies have been recently established from biopsy specimens and in two cases from circulating tumor cells (CTCs)[10–12]. Genetically engineered mouse (GEM) models have provided critical information on key mechanisms of CRPC progression and drug resistance but they do not fully recapitulate tumor heterogeneity or reflect the hierarchical organization of the tumor. Patient-derived xenografts (PDX) can reflect the heterogeneity of human tumors and are currently the most clinically relevant models[13,14]. However their feasibility remains challenged by limited tumor tissue availability and a low tumor take rate. Prostate PDX were mostly derived from either localized prostate cancer or early stage of metastatic disease. Few PDX were generated from post-treatment biopsies, which are expected to be useful for understanding intra-tumor heterogeneity and clonal evolution that emerge under the selective treatment pressure.

CTCs are derived from the primary tumor and/or metastatic sites and can be found in the blood of a proportion of patients with prostate cancer depending on their clinical stage and the applied CTC detection technology. Using the CellSearch technology (Menarini Silicon Biosystems), the detection of >5 CTCs in 7.5 ml blood has been validated as a prognostic marker for CRPC[15] and CTCs were applied as a pharmacodynamic biomarker in CRPC patients receiving chemotherapy or AR-directed therapies[16]. Additionally, CTCs were used as a non-invasive liquid biopsy to identify oncogene status in CRPC and predictive biomarkers of drug sensitivity[17,18]. Interestingly, CTCs potentially provide more comprehensive molecular information on metastatic cancer than a single metastatic lesion as they may better represent tumor heterogeneity of different metastatic foci. Because of CTC rarity in small blood volumes, diagnostic leukapheresis (DLA) is currently being investigated to obtain higher numbers of CTCs and a liquid biopsy more representative of

tumor heterogeneity[11,19–21]. Recently, a minority of CTCs with cancer stem cell features and tumorigenic activity in immuno-compromised mice has been reported to have high relevance for metastatic progression[22]. The generation of CTC-derived eXplant (CDX) models from a readily accessible blood draw at relevant time-points during disease progression can overcome some of the limitations of existing models and offers the opportunity to explore the tumorigenicity of CTCs as well as new targeting strategies. Nevertheless, it is worth noting that CDX development remains challenging owing to CTCs scarcity and technical hurdles related to their enrichment strategies. In metastatic breast cancer, Baccelli et al.[22] reported for the first time a subpopulation of CTCs with a tumor-initiating CD45−EpCAM+CD44+CD47+ cMet+ phenotype. The feasibility of establishing CDXs amenable to pharmacology-based studies has been reported in small-cell lung cancer and melanoma[23,24]. Establishment of one NSCLC CDX has also been reported[25]. In addition, ex vivo expansion of viable CTCs was successful with the establishment of permanent in vitro CTC-derived cell lines in colon cancer and the cultures of breast CTCs for drug–response testing[26,27]. Here, we report the first prostate CDX and show that this model harbored phenotypical and genetic characteristics of CRPC-NE. Comprehensive analysis of primary tumor (PT) specimens, CTCs, and the CDX and an in vitro CDX-derived cell line provide insight into the genetic basis of the tumorigenicy of CTCs and the stepwise transformation of CRPC into CRPC-NE in this unique experimental model.

## Results

**Prostate CDX establishment from a DLA product.** We first tried to establish CDX using blood samples from CRPC patients with advanced disease. Thirty milliliters blood samples were collected from 15 patients with advanced CRPC and the hematopoietic blood-cell depleted fraction was implanted in NSG mice. The number of implanted epithelial (EpCAM+ cytokeratins+) CTCs (median 230, range 0–18,389) was estimated in paired 7.5 ml blood samples processed by CellSearch (Supplementary Fig. 1A). No palpable tumor was detected within 10 months of cell implantation. We then used DLA products that were generated as part of a European FP7 prospective multi-center study (CTCTrap) aimed to evaluate increased CTC yield for genomic tumor characterization and ultimately therapy guidance. In our center, DLA products were processed from seven mCRPC patients without any noticeable side effects. Clinical characteristics are presented in Table 1. The number of CellSearch CTCs was measured in 7.5 ml blood before starting the DLA procedure and in $200 \times 10^6$ mononuclear cells of the DLA product as previously reported[20] (Table 1, Supplementary Fig. 1B). The numbers of CellSearch CTCs implanted in NSG mice after hematopoietic blood-cell depletion were extrapolated from DLA CTC counts, and ranged from 0 to 19,988 (median 698). We detected a palpable tumor within 165 days after implantation of 19,988 CTCs from Patient 3 with a doubling time of almost 6 days (Fig. 1a). Patient 3 clinical history is summarized in the Supplementary Fig. 2. In April 2014, Patient 3 had six biopsies of primary prostate tumor leading the diagnosis of metastatic adenocarcinoma with a Gleason score of 9. In July 2014, he underwent transurethral resections of prostate (TURP). Between April 2014 and May 2016, he received five lines of treatment including ADT, cabazitaxel, docetaxel, the AR inhibitor enzalutamide, and again docetaxel. The DLA was performed in April 2016 at disease progression on enzalutamide.

At the first CDX passage, FISH testing of Alu sequences confirmed the human origin of the tumor. Tumor fragments were used to propagate the CDX in successive generations of NSG

**Table 1 Characteristics of mCRPC patients and apheresis products.**

| Patients | Histology | Gleason score[a] | PSA[b] | Pre-apheresis treatments | | | | CTCs/7.5 ml blood | Apheresis | | |
|---|---|---|---|---|---|---|---|---|---|---|---|
| | | | | Chemotherapy | Castration | Enzalutamide | Abiraterone | | CTCs/200 × 10^6 cells[c] | Total CTCs[d] | CTCs xenografted[e] |
| P1 | Carcinoma | 6 (3+3) | 53 | Yes | Yes | Yes | No | 81 | 257 | 5184 | 2160 |
| P2 | Carcinoma | 7 | 562 | Yes | Yes | No | Yes | 247 | 1296 | 19,047 | 4716 |
| P3 | Adenocarcinoma | 9 (4+5) | 592 | Yes | Yes | Yes | No | 129 | 761 | 26,010 | 19,988 |
| P4 | Adenocarcinoma | – | 8 | No | Yes | Yes | No | 1 | 0 | 0 | 0 |
| P5 | Adenocarcinoma | 8 (4+4) | 40 | No | Yes | No | No | 0 | 0 | 0 | 0 |
| P6 | Adenocarcinoma | 7 (4+3) | 72 | No | Yes | Yes | Yes | 6 | 44 | 1125 | 698 |
| P7 | Adenocarcinoma | 8 (4+4) | 6 | No | Yes | Yes | No | 11 | 3 | 47 | 16 |

mCRPC: metastatic castration-resistant prostate cancer, PSA: prostate-specific antigen, CTC: circulating tumor cell.
[a]At diagnosis. [b]At the time of apheresis (ng/ml). [c]Number of CTC per 200 × 10^6 cells of apheresis product according to Andree et al.[20]. [d]Extrapolation of the total number of CTC in complete apheresis product based on CellSearch count. [e]Extrapolation of the number of CTCs xenografted based on CellSearch counts.

mice. A testosterone supplement was used until passage 7. At passage 7, testosterone dependence was tested by comparing tumor growth in groups of animals supplemented or not with testosterone. The CDX growth was similar in the presence or absence of testosterone and testosterone was thus discontinued. Histological examination showed patterns of prostate adenocarcinoma of luminal phenotype in primary tumor (PT) biopsies, whereas the aspect was that of a poorly differentiated carcinoma in the CDX, characterized by loss of glandular architecture (Fig. 1b). Immunohistochemistry (IHC) showed that, in CDX and two PTs, tumor cells were positive for EpCAM and CK8/18 and negative for CK7 and vimentin. While tumor cells in PT strongly expressed PSA and AR, those in CDX tumor were negative for both proteins. As commonly observed, a few foci of neuroendocrine cells expressing NSE, chromogranin A, and synaptophysin were detected in PT. In contrast, virtually all CDX tumor cells expressed neuron-specific enolase (NSE), chromogranin A, Ki67, and CD44 evidencing emergence of an AR-null, neuroendocrine-positive phenotype. At passage 2, CDX tumors were dissociated and human tumor cells were cultured in vitro in normal-serum-containing medium and under adherent conditions. CDX-derived cells proliferated in vitro and grew by forming both an adherent monolayer and clusters resembling microspheres (Fig. 1c). At 3 months of culture, a reference stock of the CDX-derived cell line was frozen. A permanent cell line was established with a doubling time of about 4 days. The CDX-derived cells have been subcultured over 18 months. Similarly to the CDX, the CDX-derived cell line expressed an epithelial phenotype associated with the expression of neuroendocrine markers (Fig. 1b, d). Interestingly the cell line also expressed cancer stem cell markers including CD133 and had ALDH activity (Fig. 1d). The CDX-derived cell line was tumorigenic in both NSG and nude mice with tumors that develop between 90 and 110 days of implantation, respectively.

To further characterize the CDX phenotype, we performed RNA sequencing of the prostate adenocarcinoma LNCaP cell line, the CDX and CDX-derived cell line. Unsupervised hierarchical clustering of the 1000 most variant genes identified two clusters, the first composed of LNCaP samples and the second of the CDX and the cell line (Supplementary Fig. 3). Then, we focused on 250 functional genes that are relevant for CRPC-NE progression and significantly deregulated (CPM ≥2 in at least three samples)[7,28]. These data further confirmed the two clusters and similarity of the transcriptional profiles of the CDX and the cell line (Fig. 2a, Supplementary Data 1). By supervised analysis of genes differentially expressed between LNCaP cells and the CDX, genes involved in neuroendocrine differentiation (NED) signaling pathways including E2F transcription factors and Wnt were significantly upregulated (q values ≤0.1) in the CDX while AR and Notch pathways were downregulated compared to the

LNCaP cell line (Fig. 2b). Genes implicated in neural development (Fig. 2b) and CHGA and SYP genes coding for neuroendocrine chromogranin A and synaptophysin markers respectively were overexpressed in the CDX and the CDX-derived cell line compared to LNCaP (Fig. 2c). Transcriptional regulators including STAT3 (signal transducer and activator of transcription 3), ASCL1 (Achaete-Scute family BHLH transcription factor 1), SOX2 (sex determining region Y-box 2), POU3F2 (POU class 3 homeobox 2), FOXA2 (Forkhead box A2), FOXA1 (Forkhead box A1), PDX1 (pancreatic-duodenal homebox factor 1), and REST (RE1-silencing transcription factor) as well as EZH2 (histone methyltransferase enhancer of zeste homolog 2) and TIMP-1 (TIMP metallopeptidase inhibitor 1) genes were deregulated (Fig. 2c). TP53, RB1 PTEN, and CYLD (CYLD lysine 63 deubiquitinase) tumor suppressor genes were also underexpressed. Overall, transcriptional profiling showed the deregulation of multiple genes and signaling pathways that are hallmarks of CRPC-NE progression and/or drivers of NED together with decreased AR signaling.

**Comparative genomic analysis of PT, CTCs, and the CDX.** To determine to what extent the CDX was representative of the primary tumor, we performed whole-exome sequencing (WES) of six FFPE PT biopsies performed at diagnosis, two FFPE TURP specimens, CTCs from the DLA product, and the CDX and CDX-derived cell line. Due to the lower quality of collected material, biopsies 1 and 4 were excluded from variant identification but conserved for detecting variants found in other PT specimens. WES was performed on six pools of five CTCs that were isolated from the depleted hematopoietic blood-cell fraction of the DLA product by fluorescence activated cell-sorting (FACS) (Supplementary Fig. 4). Statistics of coverage, depth of sequencing and numbers of variants identified in PT specimens, and the CDX and the CDX-derived cell line are shown in Supplementary Table 3. Statistics of coverage, depth of sequencing, allele drop out (ADO), and false-positive rate (FPR) of CTC samples are shown in Supplementary Table 4 and Supplementary Figs. 5A, B. CTC pools exhibited FPR values ranging from 7 per Mb to 21 per Mb. Principal component analysis (PCA) revealed the mutational similarity (clustering) of PT, CTC samples, and the CDX and CDX-derived cell line (Supplementary Fig. 6). Two hundred and five mutations were detected in the eight PT specimens. Among these 205 mutations, 153 (75%) were detected in only one PT biopsy, illustrating the great mutational heterogeneity of the primary tumor in this patient (Fig. 3a). Thirty-two (16%) of these 205 mutations were found in the CDX and CDX-derived cell line (Fig. 3b, c). The overlap of mutations between PT specimens and the CDX varied between 5% and 30% (Fig. 3d). These results indicate that PT specimens contained a relatively similar

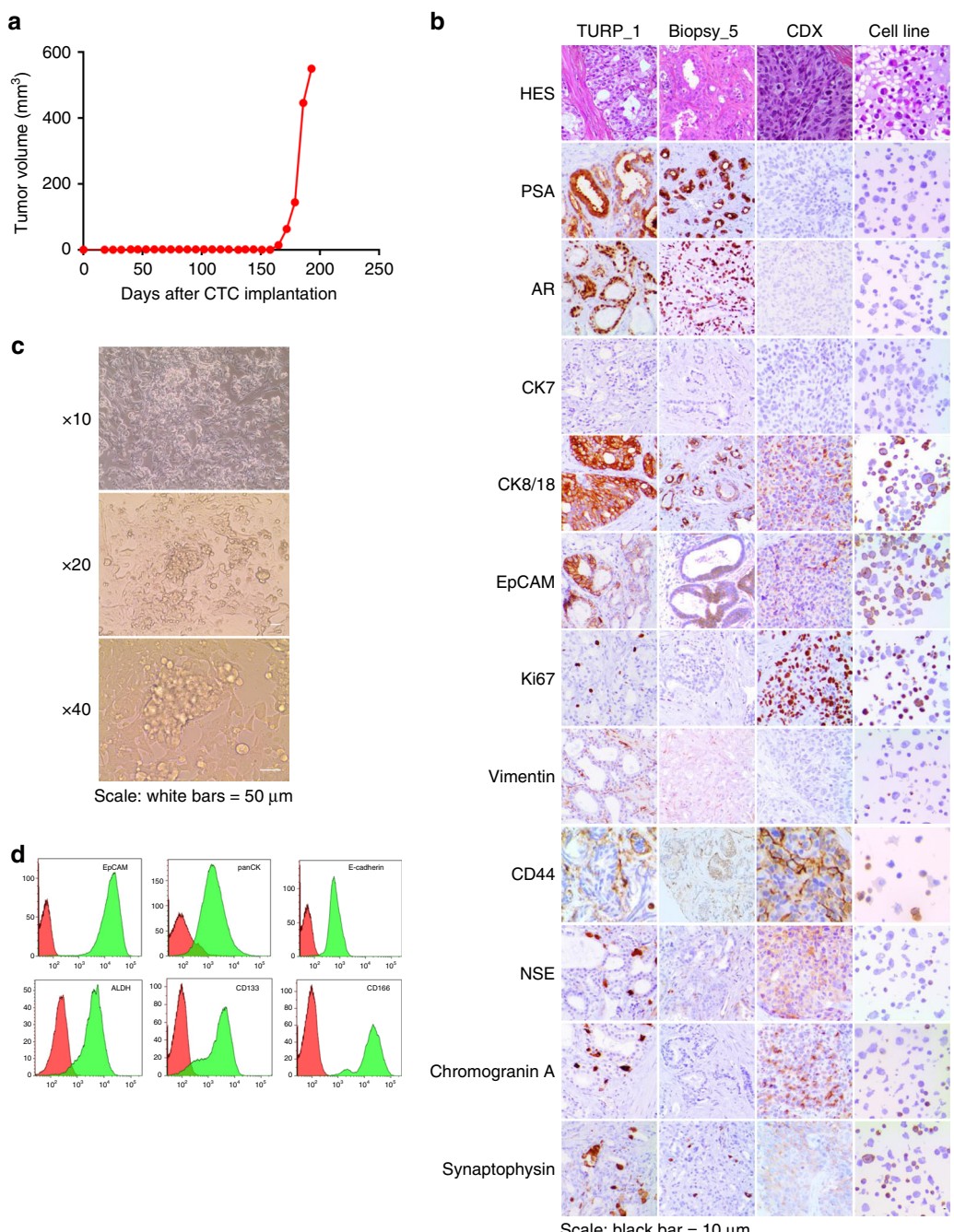

**Fig. 1 Establishment and phenotypic characterization of the CDX and the CDX-derived cell line. a** Tumor growth curve of the CDX showing the tumor volume (mm³) according to the number of days after CTC implantation. **b** Immunohistochemical characterization of PT specimens, the CDX, and the CDX-derived cell line. Representative images of HES, PSA, AR, CK7, CK8/18, EpCAM, Ki67, Vimentin, CD44, NSE, Chromogranin A, and Synaptophysin stainings of TURP_1, Biopsy_5, the CDX, and the CDX-derived cell line are shown at ×20 magnification. Scale bar represents 10 μm. **c** Representative images of the CDX-derived cell line at ×10, ×20, and ×40 magnification. Cells were growing by forming an adherent monolayer and microspheres. **d** Phenotypic characterization of the CDX-derived cell line by flow cytometry. Expression of epithelial markers, including EpCAM, pan-cytokeratins, and E-cadherin, of CD133 and CD166 cancer stem cell markers and ALDH activity are shown.

proportion of CDX mutations including a G279E TP53 mutation. All identified mutations are described in the Supplementary Data 2.

Overall, these data show the mutational heterogeneity of PT specimens and high similarity of the CDX and CDX-derived cell line.

Next, we examined shared copy-number alterations (CNAs) between PT specimens, CTCs, the CDX and the CDX-derived cell line. In contrast to mutations, only six CNA were detected in PT

specimens of which five were conserved in all CTC samples, the CDX and the cell line (Fig. 3e, f). These six CNAs were detected in most PT specimens. Five of six CNAs were conserved during disease evolution and found in tumorigenic CTCs. We observed that the TMPRSS2-ERG fusion and the NKX3-1 loss were present in all PT specimens, CTCs, the CDX and the cell line. The chromosomal segment 17p12-tel containing the TP53 and MAP2K4 genes was exclusively lost in biopsy 5 and was conserved in CTCs, the CDX and CDX-derived cell line, suggesting that

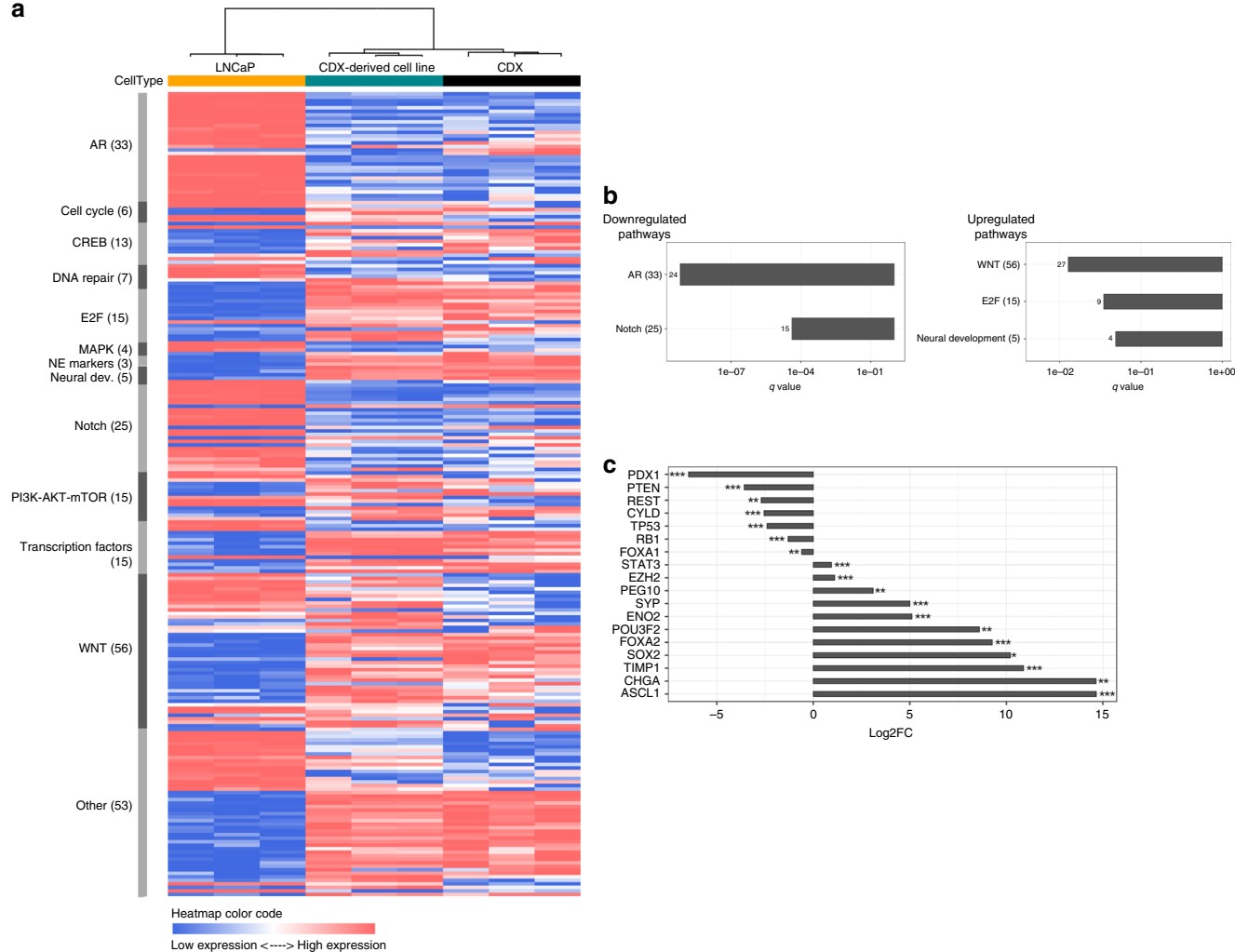

**Fig. 2 Transcriptional profile of the CDX and the CDX-derived cell line. a** Unsupervised hierarchical clustering of transcriptional profiles of the LNCaP cell line and the CDX and CDX-derived cell line. The rows show the normalized expression of 250 functional genes that are relevant for CRPC-NE progression and/or NED signaling pathways and significantly deregulated (CPM ≥ 2 in at least three samples). The number of genes analyzed per pathway is indicated in parentheses (**b**). Results of the supervised analysis of signaling pathways involved in CRPC-NE progression and/or NED that are differentially expressed between LNCaP cells and the CDX. Histogram bars represent downregulated or upregulated pathways according to their *q* value (≤0.1). The number of genes significantly deregulated in each pathway is mentioned. **c** Results of the supervised analysis of the main genes involved in CRPC-NE progression and/or NED that are differentially expressed between LNCaP cells and the CDX. Histogram bars represent underexpressed and overexpressed genes according to the fold change. \**q* value < 0.1, \*\**q* value < 0.01, \*\*\**q* value < 0.001.

17p12-tel loss can confer a selective advantage for therapy resistance and tumorigenicity of CTCs. Additionally, a smaller segment in chromosome 17 (17p12–13) which also includes *TP53* and *MAP2K4* genes was lost in four other PT specimens including biopsies 1 and 4, and TURP 1 and 2. An example of CNA profiles in a PT specimen and the CDX and CDX-derived cell line is shown in Supplementary Fig. 7.

**Genomic analysis of CTCs**. Using the criteria adopted for calling variants (present in the primary tumor and/or the CDX and/or two or more CTC samples), a set of 62 high-confidence somatic variants were identified in the six CTC samples (Fig. 4a, Supplementary Table 4). In all, 25/62 (40%) of high-confidence CTC variants arose from PT. Of 62, 35 (56%) of CTC variants were conserved in the CDX and associated with the tumorigenic activity of CTCs. Among these, 24/62 (39%) were issued from PT while 11/62 (18%) were not detected in PT. This result suggests that tumorigenic CTCs harboring these mutations either

represented minor subclones in the primary tumor or were derived from distinct metastatic sites. We also observed that 52/62 (84%) of high-confidence variants were present in two or more CTC samples (Fig. 4b) while 37/62 (60%) were CTC-private variants (not found in the primary tumor).

**CDX and CDX-derived cell line genomic characterization**. The CDX contained 80 mutations, of which 32 (40%) were issued from PT specimens (Fig. 5a) and represented only 16% of PT mutations (Fig. 3b, c). Among these 80 CDX mutations, 11 (14%) were detected in CTCs but not in PT, and were possibly arising from minor subclones in PT, or distinct metastatic sites, as mentioned above. The status (mutated/non mutated) and amino acid variation of these 11 genes in the different samples are presented in Fig. 5b. Of 80 mutations, 37 (46%) were exclusively detected in the CDX and the CDX-derived cell line (Fig. 5a) and were likely generated during the CDX in vivo development. A total of 41 CNAs were detected in the CDX, of which only

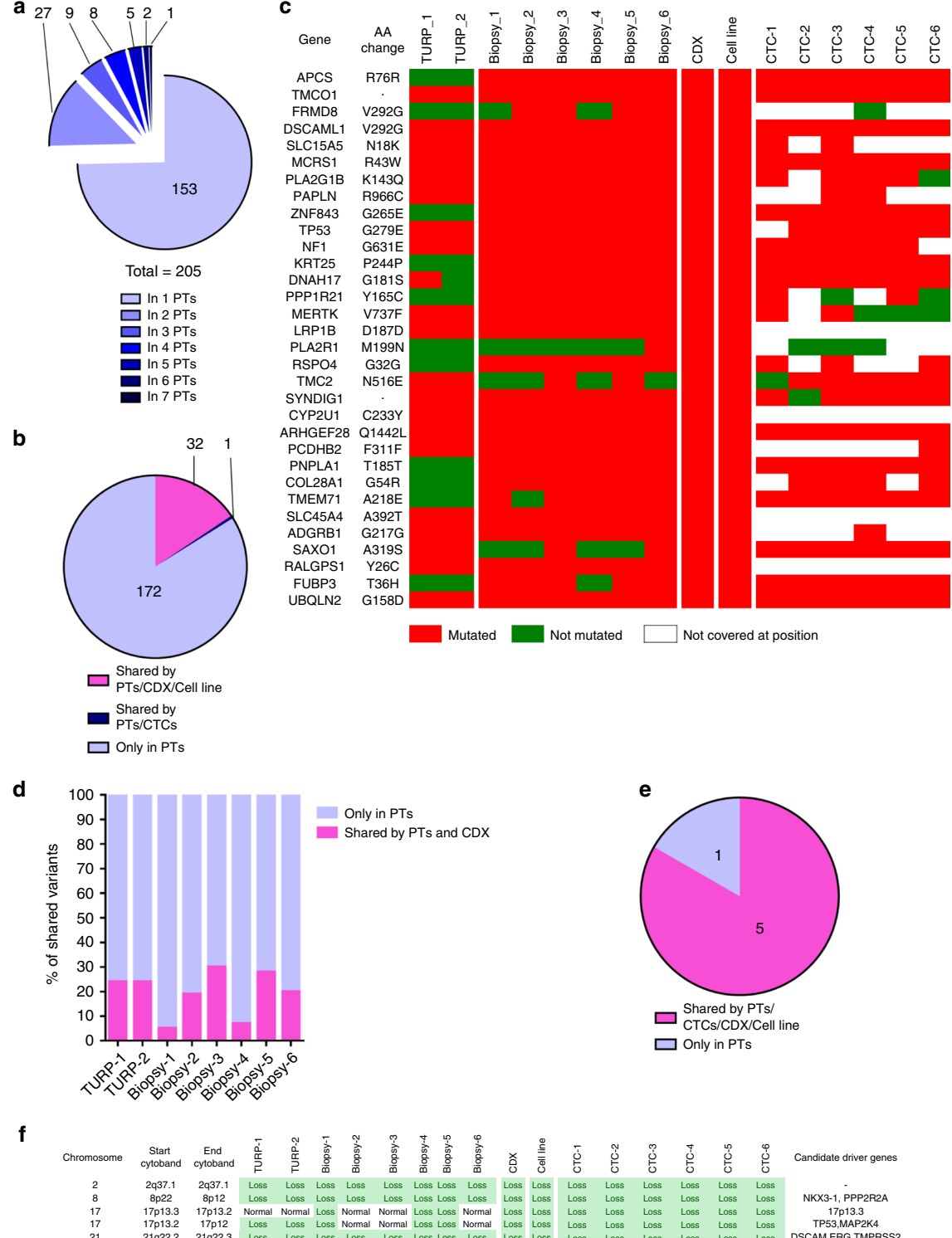

5 (12%) were issued from PT including loss of *TP53* (chromosomal segment 17p12-tel) (Fig. 5c). Nine of 41 (22%) CNAs of the CDX including *PTEN*, *RB1*, *BRCA2*, *FOXO1*, *HDAC4*, and *REST* loss were acquired in CTCs during disease evolution, suggesting that these CNAs could arise from distinct metastatic sites (Fig. 5d). Interestingly these nine CNAs were highly conserved in CTC samples. Of 41 CNAs, 27 (66%) were exclusively detected in the CDX and the cell line and were likely generated during the CDX in vivo development. A high number of CNAs as well as whole-genome doubling (WGD) were observed in the CDX and

the cell line in comparison to the low number of CNAs and diploidy of PT (Supplementary Fig. 7) consistent with genomic instability acquired during metastatic process. No bi-allelic alteration (i.e. bi-allelic deletion and/or mutation) of DNA repair pathways genes such *BRCA1, BRCA2, ATM, CDK12, RAD51, PALB2, FANCA, CHEK2, MLH1, MSH2, MLH3*, and *MSH6* was observed in CTCs, the CDX, or the cell line. Important similarity between the CDX and the CDX-derived cell line was observed in terms of both mutational and CNA landscapes (Figs. 3c, f and 5b, d). Overall, genomic characterization of the

**Fig. 3 Comparative genomic analysis of PTs and the CDX and the CDX-derived cell line. a** Numbers of mutations according to their recurrence in PT specimens. A total of 205 mutations were detected in PT specimens. In total, 153 (75%) mutations were detected in one PT, 27 were detected in two PTs, 9 were detected in three PTs, 8 were detected in four PTs, 5 were detected in five PTs, 2 were detected in six PTs, and one was detected in seven PTs. **b** Shared mutations between PTs, CTCs, and CDX and cell line. Of the 205 mutations detected in PTs, 172 were found only in PTs, 32 were shared between PTs, the CDX and the cell line and one mutation was shared between PTs and CTCs but not with the CDX and the cell line. **c** Heatmap of the 32 PT mutations shared with CTCs, the CDX and the cell line. Mutated genes, amino acid changes, and mutations status in the 16 tumor samples including two TURP, six biopsies, the CDX, the cell line, and six CTC samples are indicated. Red color indicates that the sample is mutated, green indicates that the sample is not mutated, and white indicates that the position is not covered with a sufficient depth of sequencing. **d** Percentage of mutations of each PT shared by the CDX. For each PT the number of mutations is considered as 100%. The percentage of shared mutations between each PTs and the CDX is represented in pink. Mutations present only in PTs are represented in blue. **e** Shared CNAs between PTs, the CDX and the cell line. Five of the six CNAs found in PTs are shared with CTCs, the CDX. **f** Heatmap of the five PT CNAs shared with CTCs, the CDX and the cell line. Chromosomes harboring CNAs, CNA status in the 16 tumor samples and candidate driver genes in chromosome segments are presented.

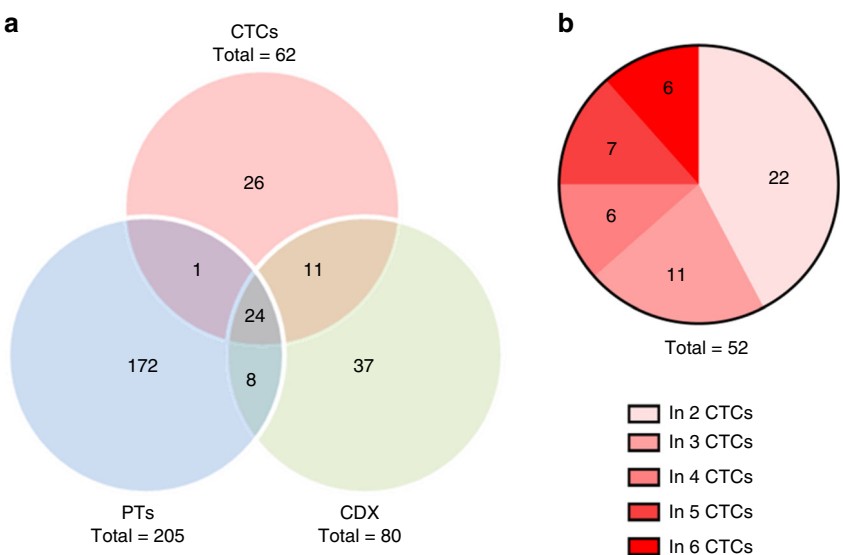

**Fig. 4 Genomic analysis of CTCs. a** Genetic relationship between CTCs, primary tumor, and the CDX. Venn diagram showing the overlap of somatic variants detected among CTCs, the primary tumor and the CDX. **b** Number of somatic variants in CTCs according to their recurrence.

CDX revealed some frequent genomic alterations found in CRPC-NE, such as *TP53* mutations, *TP53* and *RB1* loss along with *PTEN* loss which are associated with abiraterone resistance and progression towards a small-cell/neuroendocrine phenotype in GEM models[29].

**Phylogenetic relationship between PT, CTCs, and the CDX.** Twenty-one truncal alterations including the G279E-driver *TP53* mutation and three CNAs including the *TMPRSS2-ERG* fusion and *NKX3.1* loss were detected (Fig. 6). Two main branches were identified: (i) the first one composed of PT specimens was supported by nine mutations including a *FOXA1* driver mutation. As mentioned above, loss of chromosome 17 fragments including *TP53* and *MAP2K4* occurred in TURP 1, TURP 2, and biopsy 5; (ii) the second branch composed of all CTC samples, the CDX and the cell line was supported by 5 mutations and 10 CNAs including gain of *MCL1* and *MDM4* genes described as anti-apoptotic and *P53* inhibitor, respectively. Loss of cancer driver and tumor suppressor genes such as *TP53*, *MAP2K4*, *PTEN*, *RB1*, *FAT1*, *CSMD*, and *REST* was also detected in all of these samples. Two ramifications were observed, the first one composed of all CTC samples and the second of the CDX and the cell line. WGD was identified as major event occurring in the CDX and the cell line. Loss of 17p12-tel region observed in biopsy 5 including *TP53* and *MAP2K4* genes was conserved in all CTC samples, the CDX and the cell line. These data indicate that tumorigenic CTCs could derive from a minor subclone arising from biopsy 5 and harboring the largest loss of 17p12-tel

region while subclone(s) harboring loss of the 17p12–13 region found in spatially different PT regions (TURP 1 and 2) were not conserved in CTCs and the CDX.

We then examined whether genes harboring truncal or branched alterations in CTCs, the CDX and the cell line were listed in prostate cancer databases. By interrogating 2604 different prostate tumors from eight cBIOPortal studies[30,31], genes harboring the 25 truncal mutations and the *TMPRSS2-ERG* fusion were found altered. Their relative frequency varied from 0.1% to 2.4% excepted for *TP53* found at 28%, *ERG* at 30%, and *TMPRSS2* found at 10% (Supplementary Fig. 8). Eight branched mutations were found in genes with frequencies varying from 0.1% to 1.4% (Supplementary Fig. 9). Frequencies were then examined in specific prostate cancer subtypes. Truncal mutations and fusions were found in 54%, 43%, and 26% of genes altered in prostate adenocarcarcinomas, CRPC-NE, and CRPC respectively (Fig. 7a). Genes harboring branched mutations were three times more frequently altered in CRPC-NE than in adenocarcinomas (9% vs 3%) (Fig. 7b).

Genes harboring trunk CNAs were found in 1.8–6% of the 2604 prostate tumors (Supplementary Fig. 10A). The nine branched driver CNAs were found in genes with a frequency varying from 1.6% to 8% excepted for *PTEN* which was found at 18% (Supplementary Fig. 10B). Genes harboring trunk CNAs were two times more frequently altered in CRPC-NE than in adenocarcinomas (28% vs 12%) and were found in 35% of genes altered in CRPC (Fig. 7c). Genes harboring branched CNAs were found in 55%, 50%, and 37% of genes altered in CRPC,

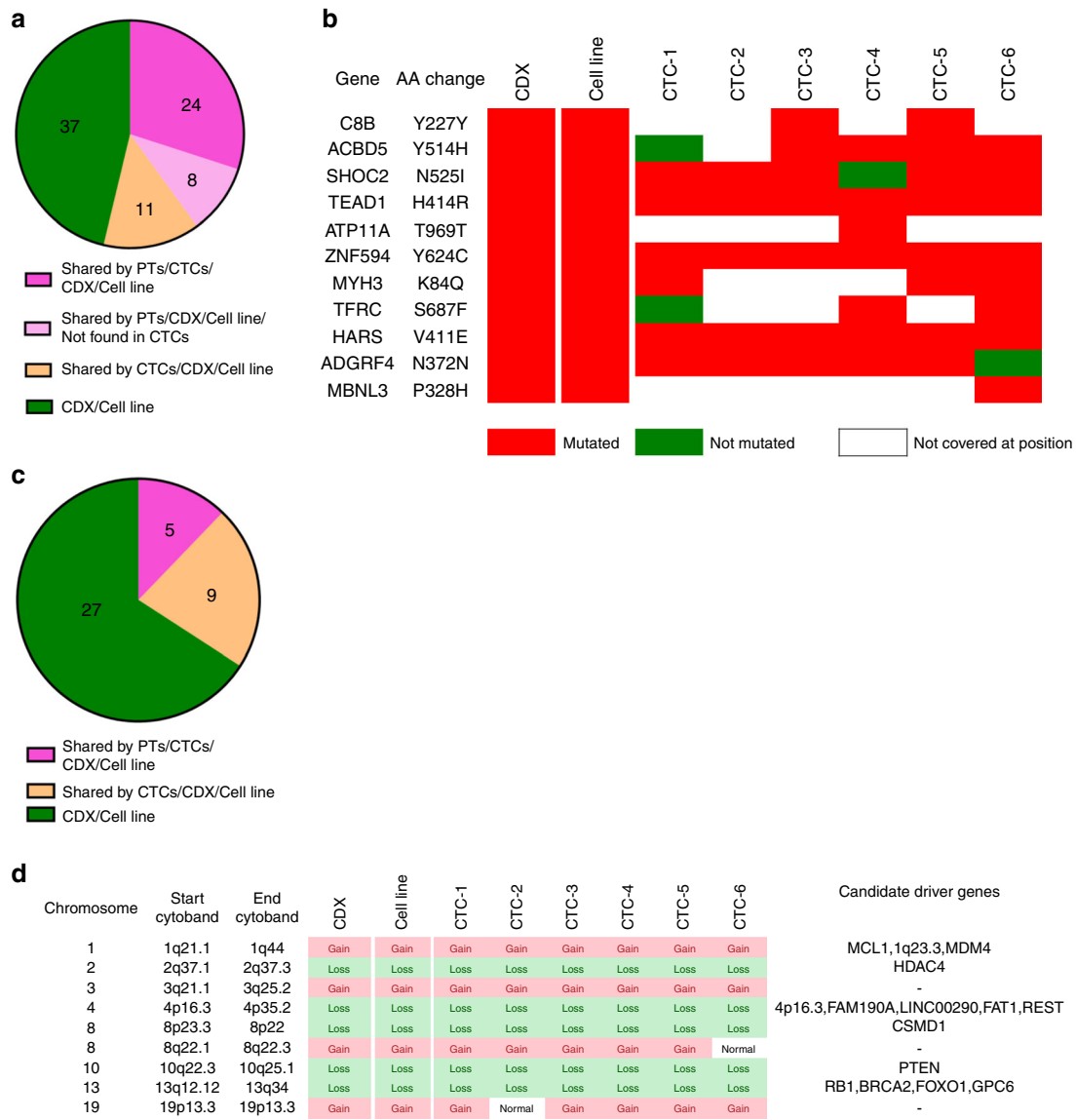

**Fig. 5 Genomic analysis of the CDX and the CDX-derived cell line. a** Origin of CDX and CDX-derived cell line mutations. Thirty-two (40%) of the 80 mutations detected in the CDX and the cell line arose from PTs. Eleven (14%) mutations arose from CTCs while 37 (46%) were exclusively found in the CDX and the cell line. **b** Heatmap of the 11 mutations of the CDX and the cell line arising from CTCs. Mutated genes, amino acid change, and mutation status of the eight tumor samples including the CDX, the cell line and six CTC samples are indicated. Red color indicates that sample is mutated, green indicates that sample is not mutated, and white indicates that the position is not covered with sufficient depth of sequencing. **c** Origin of the CDX and CDX-derived cell line CNAs. Five (12%) of the 41 CDX and cell line CNAs arose from PTs via CTCs, 9 (22%) arose from CTCs, and 27 (66%) were exclusively found in the CDX and the cell line. **d** Heatmap of the nine CDX and CDX-derived cell line CNAs arising from CTCs. Chromosome harboring the CNA, CNA status in the eight tumor samples, and candidate driver genes in chromosome segments are shown.

CRPC-NE, and adenocarcinomas respectively (Fig. 7d). Overall, interrogation of cBIOPortal database indicates an initial preponderance of neuroendocrine genetic abnormalities in primary tumor and acquisition of a more pronounced neuroendocrine genotype in CTCs with tumorigenic activity.

**Drug response of the CDX and CDX-derived cell line.** To further validate our model and in the perspective of using it to test new therapeutic compounds in CRPC, we tested whether the CDX was sensitive to docetaxel, enzalutamide, and PARP inhibitor olaparib. No significant difference in the CDX tumor growth between treated and control tumor was observed over time for any of the three drugs. Resistance to docetaxel and enzalutamide

mirrored patient response to standard-of-care CRPC therapies. As predicted by the absence of bi-allelic alteration of DNA repair pathways genes, the CDX was also resistant to olaparib (Fig. 8). This is in contrast to the PAC120 xenograft model derived from a primary prostate adenocarcinoma which was reported as sensitive to both docetaxel and enzalutamide[32] (Supplementary Fig. 11). As expected, the CDX-derived cell line exhibited resistance when treated with docetaxel or enzalutamide (Fig. 9). The CDX-derived cell line $IC_{50}$ of docetaxel was significantly higher than that of PC3 and LNCaP cells. Like PC3 cells, the CDX cell line was highly resistant to enzalutamide since the $IC_{50}$ was never reached in comparison with LNCaP which was sensitive. Overall, drug assays faithfully recapitulate the patient response to docetaxel and enzalutamide treatments.

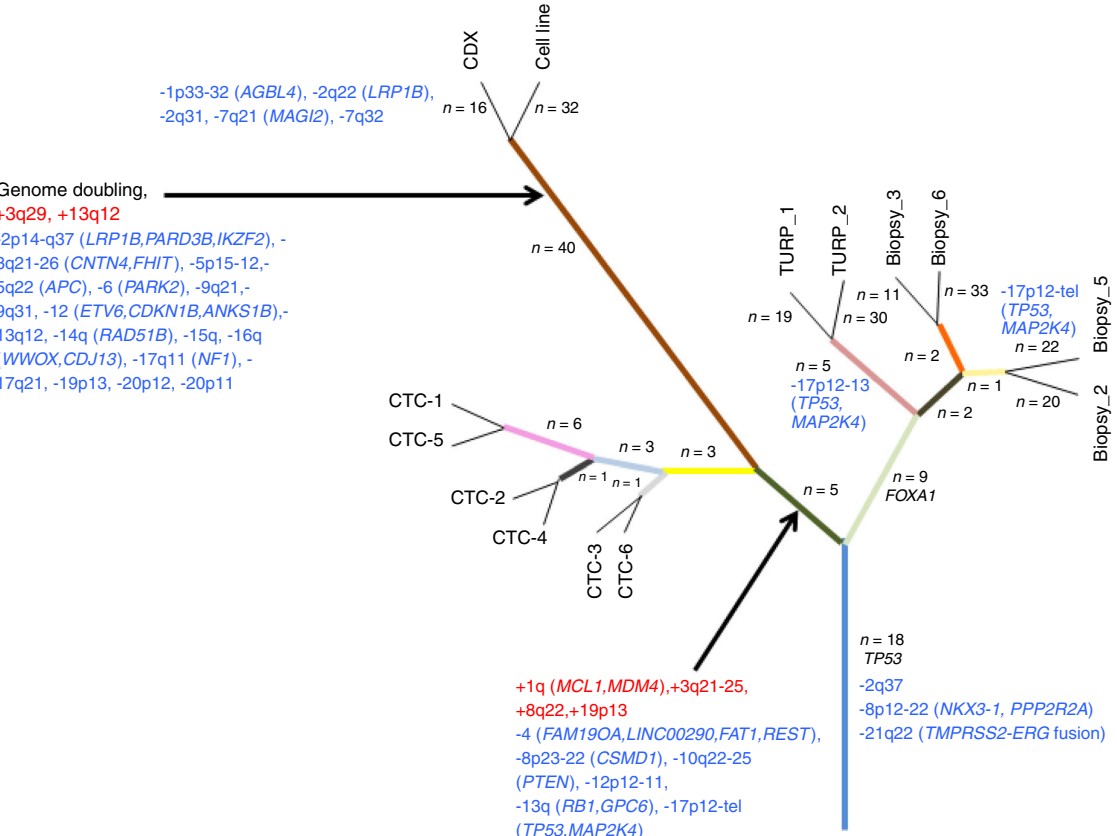

**Fig. 6 Phylogeny of the CDX and the CDX-derived cell line.** The numbers of mutations (in dark) and the CNA (loss in blue and gain in red) are mentioned on the branches of the tree. Only genes bearing driver alterations (mutations or CNAs) are indicated.

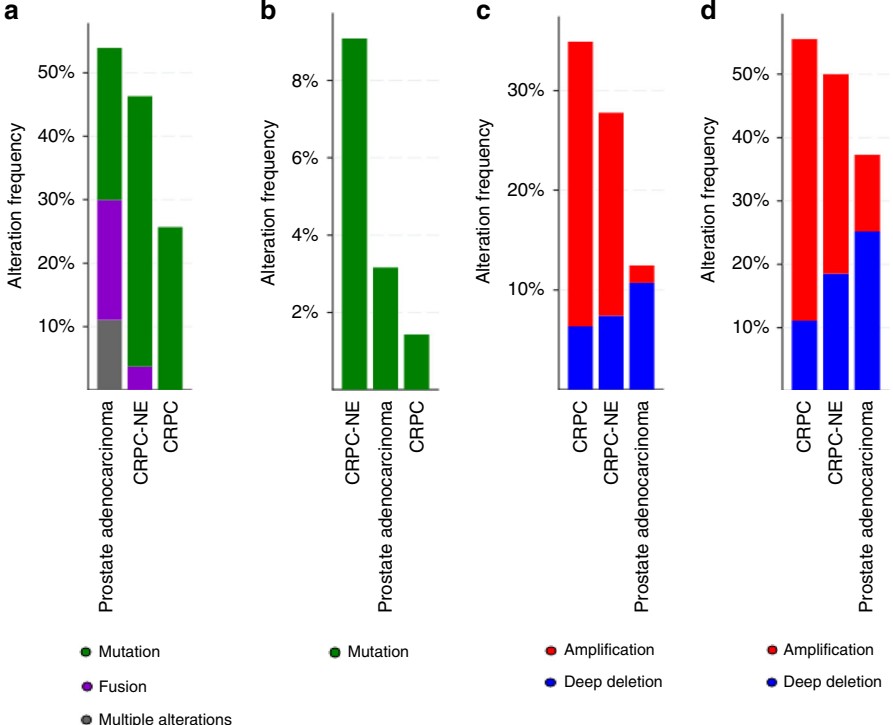

**Fig. 7 Frequency of the CDX gene alterations in prostate cancer subtypes.** The frequency of genes bearing trunk or branch alterations was examined in eight cBioPortal studies including CRPC and/or CRPC-NE tumor samples[7,64-70]. The eight interrogated studies gather 2604 tumor samples including 2029 adenocarcinoma, 70 CRPC, and 54 CRPC-NE. **a, b** Alteration frequency of genes bearing truncal (**a**) or acquired (**b**) SNVs and INDELs according to prostate cancer subtypes. **c, d** Alteration frequency of genes bearing truncal (**c**) or acquired (**d**) CNAs according to prostate cancer subtypes.

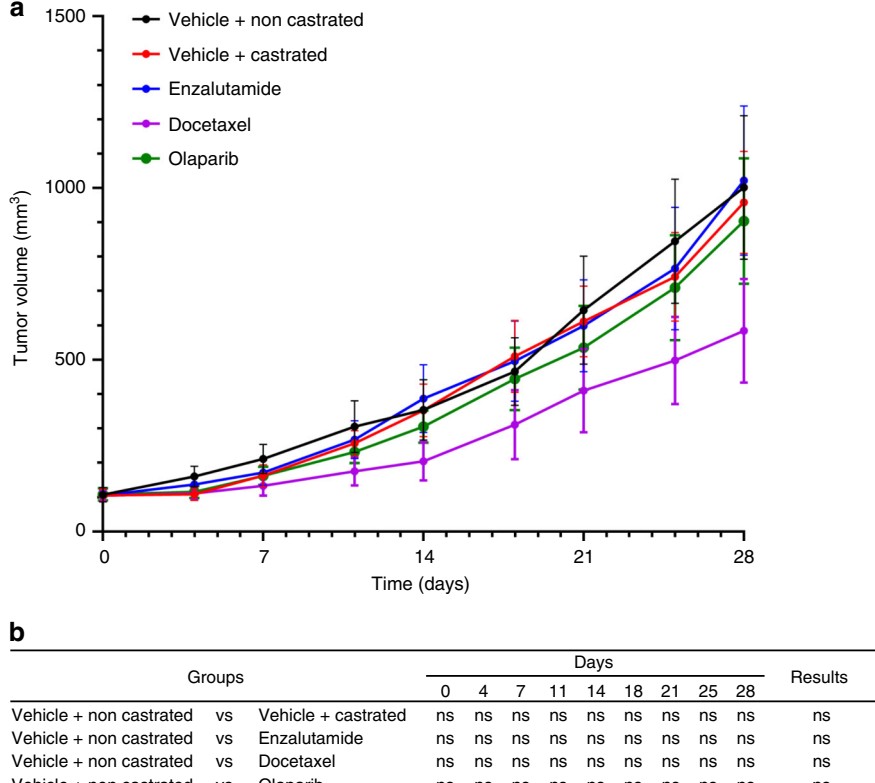

**Fig. 8 In vivo drug assays. a** The CDX is resistant to docetaxel, enzalutamide, and olaparib. Non-castrated mice bearing passage-8 CDX tumors were treated with docetaxel, enzalutamide, olaparib, or the vehicle. Control surgically castrated mice bearing CDX tumors treated by the vehicle were included. In each group n = 7. Tumor volumes of vehicle and treated groups over time after randomization are shown. Data represent mean tumor volumes ± s.e.m. **b** Comparison of the tumor volumes of vehicle and treated groups over time. A Mann–Whitney test has been applied for statistical analysis.

## Discussion

Evolution to an AR-independent cancer with features of NED is an increasingly recognized resistance mechanism to AR pathway inhibitors in a subset of patients with advanced and rapidly progressive CRPC. Largely because of the difficulty obtaining metastasis specimens and the scarcity of experimental models able to faithfully recapitulate the etiology of CRPC transformation into CRPC-NE, the molecular bases of this process are incompletely understood. Herein we report the establishment and characterization of a prostate CDX and show that this model harbor typical phenotypic and genetic features of CRPC-NE. Comprehensive analysis of primary tumor specimens, CTCs and CDX/CDX-derived cell line provided insights on the genetic basis of the tumorigenic activity of CTCs and enabled the reconstruction of the phylogenic evolution of tumorigenic CTCs. This genomic analysis suggests an order of acquisition of the key genetic drivers (i.e. *TP53, PTEN, RB1*) that govern transformation of CRPC into CRPC-NE and show that this process requires tumorigenic CTCs harboring features of CRPC-NE.

Numerous studies including our own have reported the phenotypic and genomic characterization as well as the prognostic significance of prostate CTCs[15–18]. These studies suggest a functional importance of CTCs in prostate cancer progression but until now the basis of their tumorigenicity remains unclear. After a number of failed attempts with blood samples, we thought to exploit DLA products that were generated as part of the FP7 CTCTrap project[20]. One CDX was successfully established in the case of an mCRPC patient with a high Gleason score, elevated PSA and for whom we collected the highest number of CTCs. Notably, the patient was resistant to several lines of treatment including ADT and CTCs were collected at resistance to

enzalutamide. Establishing CDX is challenging with only few models reported to date, mostly in very aggressive tumors such as small-cell lung cancer or melanoma, and always from standard blood samples[22–25]. Here, we report the use of DLA for CDX establishment, a result which suggests that DLA-increased CTC yield may enhance the chances of CDX success. Initiation and propagation of human prostate carcinoma explants is problematic and may reflect a low tumorigenic potential of prostate cancer cells whose underlying biological underpinnings are unknown. The best known models of prostate small-cell neuroendocrine carcinoma are the PC3 cell line and the LuCaP49 PDX. Initially described as a poorly differentiated adenocarcinoma, PC3 was further demonstrated to express typical features of prostatic small-cell neuroendocrine carcinoma including AR and PSA absence, neuroendocrine and CD44 marker expression, and androgen independency[33,34]. Derived from an omental fat metastasis of a prostate carcinoma exhibiting a major small-cell/neuroendocrine component, the LuCaP49 PDX model lacks expression of PSA and AR and is characterized by insensitivity to androgen deprivation and rapid tumor growth[14]. The CDX also expresses a neuroendocrine phenotype positive for synaptophysin, chromogranin, NSE, and absence of PSA and AR, and reflects the functional state of neuroendocrine prostate carcinoma in being unresponsive to androgen deprivation. The CDX is resistant to enzalutamide and docetaxel and mirrors the patient response to treatments. Although our findings may open up perspectives for developing tumor models from CTCs, they are not fit to confirm that the use of the DLA approach fully circumvents the challenges implied by CTC scarcity. We also report the establishment of an in vitro CDX-derived cell line that conserves the phenotypic, genetic, functional characteristics, and

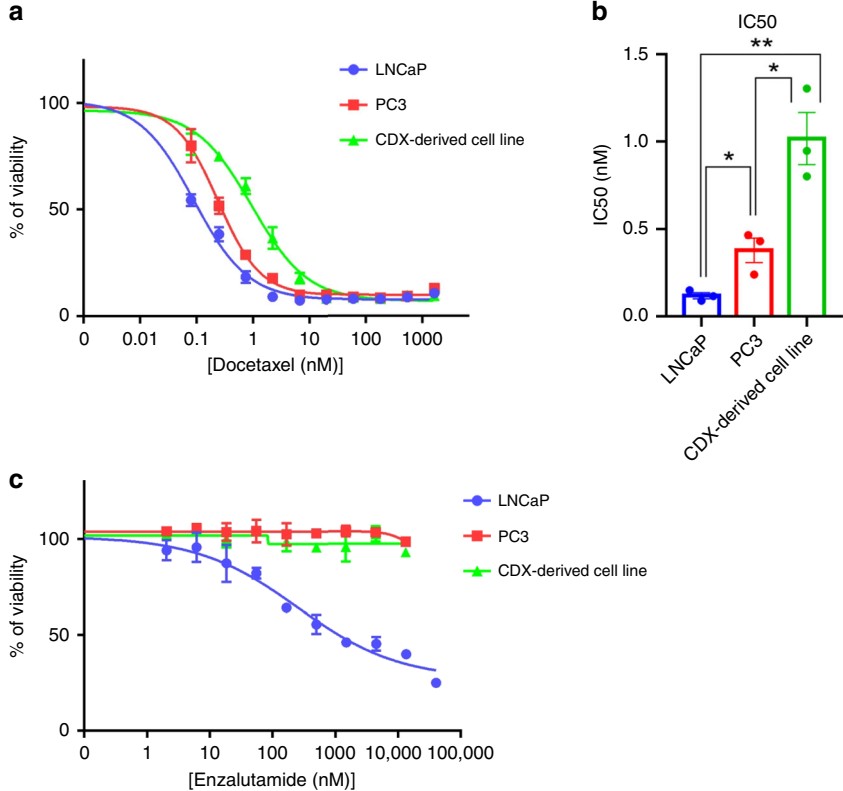

**Fig. 9 In vitro drug assays. a** Representative experiment of dose response curves of the CDX-derived cell line, LNCaP, and PC3 cell lines to docetaxel. IC$_{50}$ were 0.09, 0.24, and 0.95 nM for LNCaP, PC3, and the CDX-derived cell line respectively. **b** Mean of IC$_{50}$ values of docetaxel for the CDX-derived cell line, LNCaP, and PC3 cell lines. Data are presented as means ± SD of at least three independent experiments. An unpaired two-tailed *t*-test has been applied for statistical analysis. **c** Dose response curves of the CDX-derived cell line, LNCaP, and PC3 cell lines to enzalutamide. IC$_{50}$ was 254.5 nM for LNCaP and was not reached for PC3 and the CDX-derived cell lines.

tumorigenicity of the CDX, and provides a valuable tool for testing/modeling novel therapies.

In accordance with recent findings, we observed significant intra-tumor mutational heterogeneity in patient primary tumor[35]. Sixteen percent of these mutations were conserved in the CDX and implicated in the tumorigenic activity of CTCs. Forty percent concordance between CTCs and primary tumor mutations was detected, which supports results reported by Lohr et al.[36] in a pioneering study conducted in metastatic CRPC. Interestingly, we also observed 39% concordance between tumorigenic CTCs and primary tumor mutations, which were possibly selected under treatment pressure. Eighteen percent of CTC mutations conserved in the CDX were undetected in the primary tumor. Although we cannot rule out that these CTCs could derive from minor subclones in the primary tumor, they may have arisen from distinct metastatic sites and resulted from tumor evolution under selective pressure of treatments.

Neuroendocrine CRPC variants which emerge from prostate adenocarcinoma relapsing from AR-axis-targeted treatments were found to share clonal origin with initial adenocarcinoma[7]. Increasing evidence today including recent data from GEM models favors a transdifferentiation process where luminal prostate epithelial cells acquire typical neuroendocrine features allowing them to resist to AR-targeted therapies[37]. Recent genomic studies and GEM models have indeed highlighted that alterations (mutations, deletions) in *TP53*, *PTEN,* and *RB1* tumor suppressor genes are major co-operating events that facilitate resistance to ADT and next-generation anti-androgen therapies through an NED process associated with rapid metastatic progression and decreased overall survival[29,37–39]. GEM models of

CRPC harboring co-inactivation of *TP53* and *PTEN* failed to respond to abiraterone through transdifferentiation into CRPC-NE[29]. *RB1* loss was found to promote metastasis formation and enhance the lineage plasticity of prostate adenocarcinoma cells that was initiated by *PTEN*, while both *RB1* and *TP53* losses were found to facilitate ADT resistance[37]. Globally data from GEM models do not reveal the precise sequence and role of each driver. In the present study, we show that PT specimens harbored typical prostate adenocarcinoma features including a luminal morphology, epithelial markers, PSA and AR expression together with clonal *TMPRSS2-ERG* fusion and *TP53* mutation, and two subclonal *TP53* losses (17p12-13 and 17p12-tel). Notably CTCs obtained at resistance to enzalutamide exclusively harbored *TP53* 17p12-tel loss and acquired *PTEN* and *RB1* losses. *PTEN* and *RB1* losses were only found in CTCs and the CDX/cell line and thus possibly contributed to the resistance mechanism to AR-targeted therapies including enzalutamide. In contrast, *TP53* loss described to accelerate prostate cancer evolution toward CRPC[38] was detected in five of the eight primary tumor specimens and conserved in all CTCs. Interestingly, whereas in PT specimens *TP53* loss was found in two configurations on chromosome 17 (i.e loss of 17p12-tel and 17p12-13), only loss of the 17p12-tel segment was conserved in CTCs, the CDX and the cell line. These data suggest that the PT subclone harboring 17p12-tel *TP53* loss may have conferred a selective advantage for CTCs in driving therapy resistance and the metastatic event leading to the CDX. In the context of *TMPRSS2–ERG* fusion, the finding that *TP53* loss precedes loss of *PTEN* and *RB1* suggests a sequence of transformation of CRPC into CRPC-NE and CTCs involvement in this process. Additional drivers of NED such as *AURKA*, *MYCN*,

*TIMP-1*, *survivin*, and *REST* have been recently reported[8,40–42]. Loss of *REST* and overexpression of *TIMP1* were found in CTCs, the CDX and the cell line. No deleterious bi-allelic mutations and/ or copy-number loss in DNA repair pathway genes reported to be mutually exclusive with CRPC-NE tumors were detected in CTCs, the CDX or the cell line[28]. By expressing typical phenotypic, genetic, and functional characteristics of CRPC tumors with AR-null neuroendocrine features, this CDX model could be a unique tool for testing new therapeutic targets including inhibitors of epigenetic reprogramming factors such as *EZH2* or *SOX2* which were found to restore AR expression and sensitivity to next-generation anti-androgen therapies. This model could also be useful for the understanding of biological consequences of transdifferentiation such as cell survival and metabolic adaptation to tumor microenvironment and could provide a tractable system for therapy testing.

A number of large-scale losses were found in the CDX and CDX-derived cell line, a phenomenon which could be attributable to WGD. Our results did not allow us to determine whether WGD is occurring in CTCs or the CDX. FISH studies conducted by our and other groups have shown that CTCs may harbor hyperploid genomes and present a high degree of chromosomal instability, but WGD has not been reported. Tumors having undergone WGD evolve sub-tetraploid genomes via an increased burden of subsequent large-scale single-copy losses[43]. This may serve as a precursor of subclonal diversification and has been reported to be associated with poor outcomes[43]. By generating genetic diversity, this phenomenon may provide cells with the necessary adaptations to survive passage in the circulation and seed metastases. Importantly, about 40% of mutations and 12% of CNAs of the CDX were already present in the primary tumor at diagnosis suggesting that these genetic aberrations may have an early role in disease progression, metastasis development, and drug resistance.

Our findings open up perspectives for developing tumor models in a disease where metastasis biopsies are rarely obtained and clinically relevant experimental models are scarce. While lineage plasticity is increasingly considered as a potential mechanism of therapeutic resistance, we show here that metastatic progression of CRPC and transdifferentiation into CRPC-NE engage tumorigenic CTCs with CRPC-NE features. This CDX model may be a unique tool to improve our understanding of the genetic and epigenetic mechanisms that drive CRPC transformation into CRPC-NE and the therapeutic approaches that may reverse or delay this process.

## Methods

**Patients, blood sampling, and DLA**. The study (IDRCB2008-A00585-50) was conducted at Gustave Roussy (Villejuif, France), authorized by the French national regulation agency ANSM (Agence Nationale de Sécurité du Medicament et des produits de santé), and approved by the Ethics Committee and our institutional review board. Patients with prostate cancer were recruited into the study between March 2014 and June 2016. Informed written consent was obtained from all patients. Blood was drawn in CellSave (Menarini Silicon Biosystems, Huntington Valley, PA, USA) and EDTA tubes before starting DLA and immediately transferred to the laboratory. DLA were performed using the Spectra Optia (Terumo BCT inc., Lakewood, CO, USA) according to the manufacturer's instructions and conditions for CTC isolation have already been reported[11,20].

**Post DLA sample handling**. White blood cells (WBC) and mononuclear cells counts were determined using a FACS Canto 2 (BD Biosciences, Franklin Lakes, NJ, USA). Samples were divided into three aliquots under sterile conditions. For CellSearch (Menarini Silicon Biosystems) analysis, an aliquot of the DLA product containing $2 \times 10^8$ WBC was diluted with CellSearch Circulating Tumor Cell Kit Dilution Buffer (Menarini Silicon Biosystems) to a final volume of 8 ml in a CellSave tube (Menarini Silicon Biosystems). The second and third aliquots of the DLA product were depleted from WBC populations using the RosetteSep CTC enrichment Cocktail containing anti-CD36 (StemCell Technologies, Vancouver, Canada) and used for CTCs and WBC isolation by FACS and for CDX

establishment, respectively. Before RosetteSep enrichment, the second and third aliquots of the DLA product were mixed to patient erythrocytes to reach a WBC/ erythrocytes ratio of 1:40. Erythrocytes were isolated by centrifugation of four 9 ml EDTA blood tubes from each patient at $800 \times g$ for 10 min. $3 \times 10^8$ WBC from the DLA product mixed to erythrocytes transferred into a CellSave tube were used[11,20] for CTCs and WBC isolation by FACS. $22 \times 10^8$ WBC mixed to erythrocytes were used for CDX establishment.

**CTC enumeration using the CellSearch platform**. Blood samples were collected on CellSave® tubes and run with CellSearch (Menarini Silicon Biosystems, Huntingdon Valley, PA) using the CTC kit (Menarini) according to the manufacturer's instructions and training. The analysis an aliquot of the DLA product containing $2 \times 10^8$ WBC was diluted to a final volume of 8 ml with CellSearch CTC Kit Dilution Buffer (Menarini) stored at room temperature (RT) and transferred into a CellSave® tube containing CellSave preservative reagent (Menarini). The sample was processed using the CellTracks Autoprep system using the CTC kit. The cartridge from the DLA product was scanned using the CellTracks analyzer II.

**CTC enrichment before implantation into mice**. Fifty microliters of the RosetteSep cocktail (StemCell Technologies) was added for each 1 ml of blood or DLA product and incubated 20 min at RT. After incubation, the sample was diluted with an equal volume of Hank's Balanced Saline Solution (HBSS) (Life Technologies, Carlsbad, CA, USA) supplemented with 2% fetal bovine serum (FBS) (Life Technologies). The solution was then carefully layered on top of 15 ml Ficoll-Paque Plus (GE-Healthcare, Little Chalfont, UK) and centrifuged for 20 min at $1200 \times g$ without brake. Enriched cells were collected, washed with 50 ml HBSS/2% FBS, and centrifuged for 5 min at $250 \times g$. Cells were resuspended in 100 µl of cold HBSS supplemented with 100 µl cold Matrigel (Corning, NY, USA) and kept on ice until implantation in mice.

**Growth of CDX in immunocompromised mice**. Before CTC implantation, NOD. Cg-*Prkdc*<sup>scid</sup> *Il2rg*<sup>tm1Wjl</sup>/SzJ mice (NSG) six-week-old male mice (Charles River Laboratories, Wilmington, MA, USA) were anesthetized by peritoneal injection of 10 mg/ml ketamine and 1 mg/ml xylazine. The upper dorsal region of mice was shorn and the skin was aseptized with a chlorhexidine solution, incised at the level of the interscapular region and CTC were injected in 200 µl HBSS/Matrigel in the interscapular fat pad. A 10 mg testosterone capsule was inserted into the left flank of the animal. Mice were monitored every day. The palpable tumor was measured once a week and the tumor volume was determined as (tumor length × tumor width²)/2. When the tumor volume reached 1770 mm³ or when mice presented signs of deteriorated health status, the tumors were aseptically excised and dissected into fragments of approximately 20 mm³. Tumor fragments were passaged into NSG mice and the remainder of the tumor was used for Alu sequence detection, IHC and molecular analysis, and cell line establishment. A 10 mg testosterone capsule was used for CDX propagation until passage 7. At passage 7, testosterone dependence was tested by comparing tumor growth in groups of animals supplemented or not with testosterone. The take rate of CDX tumors was higher by using tumor fragments rather than the CDX-derived cell line (100% vs 50%, respectively) and fragments were generally used for the CDX propagation. Mice were housed in pathogen-free animal housing at the Center for Exploration and Experimental Functional Research (CERFE, Evry, France) animal facility in individually ventilated cages (IVC) of Polysulfone (PSU) plastic (mm 213 W × 362 D × 185 H, Allentown, USA) with sterilized and dust-free bedding cobs, access to sterilized food and water ad libitum, under a light–dark cycle (14-h circadian cycle of artificial light) and controlled RT and humidity. Mice were housed in groups with a maximum of six animals during 7-day acclimation period and of a maximum of six animals during the experimental phase. The animal care, housing, and all experiments were performed in accordance with French legislation concerning the protection of laboratory animals and in accordance with a currently valid license for experiments on vertebrate animals, issued by the French Ministry of Higher Education, Research and Innovation (Ministère de l'Enseignement supérieur, de la Recherche et de l'Innovation, MESRI).

**Enrichment, detection, and isolation of CTCs and CD45⁺cells**. CTC enrichment was performed using the RosetteSep cocktail (StemCell Technologies) as described above. Enriched cells were washed with 1× PBS and centrifuged at $560 \times g$ for 5 min. The pellet was then resuspended with 100 µl of fixative solution A of Fix&Perm kit (Thermo Fisher Scientific Inc., Waltham, MA, USA), washed with 1× PBS, and centrifuged at $370 \times g$ for 5 min. The pellet was resuspended in 100 µl of permeabilization solution medium B of Fix&Perm kit and 50 µl of a staining solution of cytokeratins-PE (cytokeratins 8, 18, 19) and CD45-APC antibodies of a CellSearch CTC kit (Menarini Silicon Biosystems) and 5 µl anti-vimentin-FITC antibody (clone V9, Santa Cruz Biotechnology, Dallas, Texas, USA) was added. The cell suspension was then incubated for 20 min at RT, washed with 1× PBS and centrifuged at $370 \times g$ for 5 min. The pellet was resuspended in 300 µl of 1× PBS and kept at +4 °C. Hoechst was added before cell sorting. Individual CTC isolation was performed using a BD FACSARIA III cell sorter (BD Biosciences) equipped with four lasers (a 405 nm laser, a 488 nm laser, a 561 nm laser, and a 640 nm laser). The system was run with 20 psi pressure, a 100 µm nozzle, and the yield

precision mode. Hoechst 33342-positive elements were first gated. The second gate enabled selection of CD45-APC-negative events. CD45-APC$^-$/CK-PE$^+$/Vim-FITC$^-$ cells were sorted and collected in a 96-well plate. Two hundred control CD45-APC$^+$/CK-PE$^-$ cells were sorted in one well. Plates were centrifuged 10 min at 280 × $g$ and frozen at −20 °C for at least 30 min.

**WGA, quality control, dsDNA**. Whole-genome amplification (WGA) of CTCs and CD45-positive cells was performed using the Ampli1 WGA kit (Menarini Silicon Biosystems) according to the manufacturer's instructions. The quality of Ampli1 WGA products was checked by multiplex PCR as described by Polzer et al.[44]. To increase the total dsDNA content in Ampli1 WGA products, ssDNA molecules were converted into dsDNA molecules using the Ampli1 ReAmp/ds kit (Menarini Silicon Biosystems Inc.).

**Isolation of genomic DNA from blood, PT, CDX and cell line**. DNA from formalin-fixed paraffin-embedded tumor biopsies was purified with QIAamp DNA FFPE Tissue kit (Qiagen, Hilden, Germany) according to the manufacturer's protocol. DNA from the CDX was extracted with the AllPrep DNA/RNA kit (Qiagen) and germline DNA and cell line DNA was purified with the QIAamp DNA blood kit (Qiagen).

**Whole-exome sequencing**. Library preparation, exome capture, sequencing, and data analysis have been done by IntegraGen SA (Evry, France). Genomic DNA is captured using Agilent in-solution enrichment methodology (SureSelect SureSelect XT Clinical Reasearch Exome, Agilent) with their biotinylated oligonucleotides probes library (SureSelect XT Clinical Reasearch Exome—54 Mb, Agilent), followed by paired-end 75 bases massively parallel sequencing on Illumina HiSeq4000. For detailed explanations of the process, see Gnirke et al.[45]. Sequence capture, enrichment, and elution are performed according to the manufacturer's instruction and protocols (SureSelect, Agilent) without modification except for library preparation performed with the NEBNext Ultra kit (New England Biolabs). For library preparation, 600 ng of each genomic DNA are fragmented by sonication and purified to yield fragments of 150–200 bp. Paired-end adaptor oligonucleotides from the NEB kit are ligated on repaired, A-tailed fragments then purified and enriched by eight PCR cycles. In total, 1200 ng of these purified libraries are then hybridized to the SureSelect oligo probe capture library for 72 h. After hybridization, washing, and elution, the eluted fraction is PCR-amplified with nine cycles, purified, and quantified by QPCR to obtain sufficient DNA template for downstream applications. Each eluted-enriched DNA sample is then sequenced on an Illumina HiSeq4000 as paired-end 75b reads. Image analysis and base calling is performed using Illumina Real Time Analysis (2.7.6) with default parameters.

**Sequence alignment and variant calling**. Base calling was performed using the Real-Time Analysis software sequence pipeline (2.7.7) from Illumina with default parameters. Sequence reads from amplified DNA (CTC and CD45 pools) were trimmed for Ampli1 adapters with Cutadapt (1.14)[46]. Human reads from xenograft samples were extracted by bamcmp[47]. Reads were then aligned to the human genome build hg38/GRCh38.p7 using the Burrows-Wheeler Aligner (BWA) tool[48]. Duplicated reads were removed using Sambamba[49]. Variant calling of single-nucleotide variants (SNVs) and small insertions/deletions (indels) was performed using the Broad Institute's GATK[50,51] HaplotypeCaller GVCF tool (3.7) for germline variants and MuTect2[52] tool (2.0, --max_alt_alleles_in_normal_count = 2; --max_alt_allele_in_normal_fraction = 0.04) for somatic variants. To keep only reliable somatic variants, we then applied the following post-filtering steps:

- Variants classified as "PASS" or "t_lod_fstar" by MuTect2 (and not flagged as PID).
- Coverage ≥8 in the tumor and matched normal sample.
- QSS score ≥30.
- Variant allele fraction in the tumor (VAFT) ≥0.05 with ≥5 mutated reads, variant allele fraction in the normal sample (VAFN) = 0.

Additional criteria were applied to generate a high-confidence set of variants from CTCs. Variants had to be present in either the primary tissue (at least 1 PT specimen) or the CDX or at least one other CTC sample.

Bam-readcount (https://github.com/genome/bam-readcount) was used to rescue reliable variants that were present in at least two tumor samples and were not detected by Mutect2 because of their low VAF. Ensembl's Variant Effect Predictor (VEP, release 87)[53] was used to annotate variants with respect to functional consequences (type of mutation and prediction of the functional impact on the protein by SIFT.2.2 and PolyPhen 2.2.2) and frequencies in public (dbSNP147, 1000 Genomes phase 3, ExAC r3.0, COSMIC v79) and in-house databases. We used the Cancer Genome Interpreter[54] to predict driver and passenger mutations.

**ADO and false-positive rate estimation**. CTC and CD45$^+$ pool DNA were amplified before sequencing. To estimate ADO, we selected all reliable variants in germline or CD45 DNA using HaplotypeCaller with the following post-filtering: coverage ≥8 in both samples, ≥5 variant reads representing ≥5% of sequenced reads at that position, genotype quality ≥30. We then compared the proportions of

normal/variant reads in the germline and CD45 DNA using Fisher's exact test. Variants with a significant difference ($P < 0.05$), a variant allele fraction between 0.2 and 0.8 in germline DNA and <0.1 or >0.9 in the CD45 DNA were considered to have undergone ADO. To estimate the false-positive rates in CTC samples, we divided the number of potentially false-positive events by the number of target bases covered ≥8X in the same sample. We adopted a conservative approach and considered as false positive all events not found in the primary tumor and the CDX.

**Copy-number analysis**. To identify CNAs, we identified germline single-nucleotide polymorphisms (SNPs) in each sample and we calculated the coverage log-ratio (LRR) and B allele frequency (BAF) at each SNP site. Genomic profiles were divided into homogeneous segments by applying the circular binary segmentation algorithm, as implemented in the Bioconductor package DNAcopy[55], to both LRR and BAF values. We then used the Genome Alteration Print (GAP) method[56] to determine the ploidy of each sample, the level of contamination with normal cells, and the allele-specific copy number of each segment. Ploidy was estimated as the median copy-number across the genome. Chromosome aberrations were then defined using empirically determined thresholds as follows: gain, copy number > ploidy +0.5; loss, copy number < ploidy −0.5; high-level amplification, copy number > ploidy +2; homozygous deletion, copy number < 0.5. Finally, we considered a segment to have undergone LOH when the copy number of the minor allele was equal to 0.

**Characterization of known copy-number changes in CTCs**. As expected, the log-ratio (LRR) and BAF profiles of CTC samples were too noisy to obtain reliable pangenomic copy-number profiles. However, we observed that many chromosome segments displayed allelic imbalances consistent with the presence of chromosome aberrations identified in other samples, in particular CDX and cell line samples. We used these allelic imbalances to detect chromosome aberrations identified in other samples as follows:

(1) For each CTC and each chromosome aberration, we counted the number of SNPs with consistent (e.g. BAF > 0.5 in the CTC and tumor samples) and discordant allelic imbalance.
(2) We used Fisher's exact test to identify chromosome segments with a significant enrichment in consistent SNPs.
(3) We considered an aberration to be present in a CTC sample if the Fisher test was significant ($p$ value < 0.05) with ≥80% consistent SNPs.

**Classification of tumor samples based on mutation data**. To identify samples with similar mutational profiles, we selected all variants present in at least two samples and classified the samples based on their VAF across these mutations using PCA and hierarchical clustering (Ward method, cosine distance). This method allows regrouping samples sharing the same mutational profile. In practice, the matrix of VAF was used as input, and the PCA (resp. hierarchical clustering) was performed on this matrix using prcomp (resp. hclust) function from Bioconductor stats package.

**Phylogenetic tree**. All non-silent somatic mutations present in at least two samples were considered for the purpose of determining phylogenetic trees. Trees were built using a binary presence/absence matrix built from the VAF of each sample (present = VAF > 0). The R Bioconductor package phangorn v2.3.1[57] was used to perform the parsimony ratchet method[58], generating unrooted trees. The number of mutations and driver mutations on each branch were then determined by selecting mutations present in all samples downstream the branch.

**Cultured derived cell line establishment and cell culture**. After resection, tumor fragments were conserved in RPMI 1640 medium (Life Technologies, Carlsbad, CA, USA) and immediately transferred to the laboratory. After two washes in 1× PBS (Life Technologies) and incubation for 10 min in a 10 ml 1× PBS solution containing 1:10 penicillin/streptomycin (penicillin 10,000 units/ml, streptomycin 10,000 μg/ml; Life Technologies), the tumor fragments were first mechanically dissociated using a scalpel before enzymatic dissociation with the Tumor Dissociation Human Kit (Miltenyi Biotech, Köln, Germany) according to the manufacturer's protocol. Then cell suspension was successively filtered on the 100-μm and 40-μm cell strainer and washed with PBS 1× before counting. Depletion of mouse cells was performed with the Mouse Cell Depletion Kit (Miltenyi Biotec) according to the manufacturer's protocol using an AutoMacs Pro Separator (Miltenyi Biotec). Tumor cells were then centrifuged and resuspended in Advanced DMEM/F12 medium (Life Technologies) supplemented with 10% FBS, 1% antibiotics (penicillin–streptomycin; Life Technologies), and 1% ultraglutamine (Lonza, Basel, Switzerland). After counting, cells were plated in six-well plates (TPP, Trasadingen, Switzerland). Cells were observed three times a week and passaged in tissue culture flasks for cell expansion, freezing, or characterization; cells were detached using 0.005% trypsin-EDTA (Life Technologies) before centrifugation and counting. The same normal-serum culture medium was used for cell expansion and permanent culture. LNCaP and PC3 cell lines were obtained from the American Type culture Collection (ATCC) and grown in DMEM-Glutamax medium (Life

Technologies) supplemented with 10% FBS, 1% antibiotics (penicillin–streptomycin; Life Technologies).

**In vivo drug assays**. Sixty-five, 6–9-week-old male athymic nude mice (*Hsd: Athymic Nude-Fox1$^{nu}$*) provided by ENVIGO (Gannat, France) were implanted with passage-8 CDX. When tumor volumes ranged from 60 to 200 mm$^3$, seven mice were allocated to five groups to obtain groups with homogeneous mean and median tumor volumes. Treatments or vehicle were randomly attributed to the different groups. Animals were treated by intraperitoneal injection of 20 mg kg$^{-1}$ docetaxel (Taxotere, Sanofi-Aventis, France) dissolved in 0.9% NaCl on days 0 and 21, or per oral injection of 60 mg kg$^{-1}$ enzalutamide (915087-33-1, Medchemexpress, NJ, USA) dissolved in 0.1% Tween-80, 5% DMSO in 1% carboxymethyl cellulose in sterile deionized water, or per oral injection of 50 mg kg$^{-1}$ olaparib (ku0059436, Selleckchem, Houston, USA) dissolved in 10% DMSO, 30% w/v kleptose (HP-β-CD) in sterile water from day 0 to 41. Vehicle-treated mice received 0.1% Tween-80, 5% DMSO in 1% carboxymethyl cellulose in sterile deionized water per os from day 0 to 41. A control group of mice receiving the vehicle was surgically castrated. During the experimental period, animals were monitored every day for physical appearance, behavior, and clinical changes. For each animal, any sign of illness or any change in behavior related or not to the treatment was recorded. Tumor volume was evaluated with a caliper blinded to treatment group twice a week during the experimental period (from D0 to D39) and once a week during the follow-up period. All animals were weighed and tumor size measured at the same time. From day 28, animals whose tumor size reached 1764 mm$^3$ had to be sacrificed for ethical reasons. Control experiments were performed using the PAC120 xenograft model[32] according to the same protocol. Statistical analysis was performed using GraphPrism5 software. Mann–Whitney test has been applied to compare the tumor volumes of vehicle and treated-groups over time.

**In vitro drug assays**. The CDX-derived cell line, PC3, and LNCaP cells were seeded in quadruplicate into 384-well plates. After 24 h incubation, cells were treated with docetaxel (Taxotere, Sanofi-Aventis, France) and enzalutamide (915087-33-1, Medchemexpress, NJ, USA) for 5 days. Docetaxel and enzalutamide were prepared at 10 mM in DMSO. Drugs were finally diluted in Advanced DMEM medium (Life Technologies). Cell viability assays were performed using CellTiter-Glo Luminescent Cell Viability assay kit (Promega, Madison, Wisconsin, USA). Luminescence was measured by Victor X4 Series Multilabel Plate Readers (Perkin Elmer, Waltham, MA, USA). The generation of drug–response curves and determination of IC$_{50}$ values were achieved using Prism software. Data are presented as means ± SD of at least three independent experiments. Statistical analysis was performed using GraphPrism5 software. Unpaired two-tailed *t*-test has been applied to compare the IC$_{50}$ of docetaxel between the three cell lines.

**Alu hybridization**. Alu-FISH was performed using an Alu positive control probe (Ventana Medical Systems; Roche Diagnostics, Rotkreuz, Switzerland) on Ventana Discovery xT instrument with respective reagents (Roche Diagnostics). Briefly, dewaxed and rehydrated paraffin sections were pretreated with RipoPrep (32 min, 37 °C) and RipoClear (12 mn, 37 °C) followed by the protease 3 treatment for 20 min. The Alu probe was hybridized for 1 h at 50 °C. Sections were washed two times with RiboWash (2× SSC) for 8 min at 45 °C, then treated by RiboFix (20 min, 37 °C). After incubation with an anti-FITC Biotin SP-conjugated antibody (dilution 1:200; Jackson ImmunoResearch Laboratories), chromogenic revelation was performed with Blue-Map kit for 1 h, and then nuclear Red Stain II (Ventana Medical Systems, Roche Diagnostics, Rotkreuz, Switzerland) counterstaining was performed for 8 min.

**Immunohistochemistry**. IHC was performed on FFPE tissue from patient's primary tumor samples, the CDX and the cell line using with anti-PSA, AR, CK7, CK8/18, EpCAM, Ki67 antigen, Vimentin, CD44, NSE, Chromogranin A, and Synaptophysin antibodies (Supplementary Table 1). IHC was performed manually using the coverplate system. Two protocols were developed depending of the species used to raise primary antibodies (mouse or rabbit) to avoid background due to mouse cells present in the CDX. Antigen retrieval was performed by pH8 EDTA buffer incubation for 40 or 60 min at 98 °C (Supplementary Table 1). Mouse Klear kit (GBI labs) and EnVision rabbit kit (Agilent Technologies, Dako) were used for primary mouse and rabbit antibodies, respectively. Primary antibodies were incubated at RT for 1 h. IHC stains were examined by an experienced pathologist (J.Y.S.).

**Immunofluorescence and FACS analysis**. Aldehyde dehydrogenase (ALDH) activity was performed using the Aldefluor kit (StemCell Technologies, #01700) according to the manufacturer's instructions. EpCAM, CD133, CD166, pan-cyto-keratins, and E-cadherin antibodies were used according to the manufacturer's protocol. Fixation and permeabilization steps were performed for pan-cytokeratins and E-cadherin antibodies using the Fix&Perm kit (Thermo Fisher) according to the manufacturer's instructions. 2 × 10$^5$ cells were incubated with each antibody or corresponding negative control isotype antibodies (Supplementary Table 2) at RT for 20 min. Acquisition was performed with a LSR Fortessa cytometer (BD

Biosciences) equipped with BD FACS Diva software. Data were analyzed using the Kaluza software (Beckman Coulter).

**RNA extraction and sequencing**. RNA was extracted from the LNCaP cell line, the CDX and the CDX-derived cell line for RNA sequencing using the AllPrep DNA/RNA Mini Kit according to the manufacturer's instructions (Qiagen). Library preparation, sequencing, and data analysis have been done by IntegraGen SA (Evry, France). Libraries were prepared with NEBNext Ultra™ II Directional RNA Library Prep Kit for Illumina protocol according to the supplier recommendations.

Briefly, the key stages of this protocol were successively, the purification of PolyA-containing mRNA molecules using poly-T oligo attached magnetic beads from 100 ng total RNA (with the Magnetic mRNA Isolation Kit from NEB), a fragmentation using divalent cations under elevated temperature to obtain approximately 300 bp pieces, double-strand cDNA synthesis and finally Illumina adapters ligation and cDNA library amplification by PCR for sequencing. Sequencing was then carried out on Paired-End 100b reads of Illumina NovaSeq. Image analysis and base calling were performed using Illumina Real Time Analysis (3.4.4) with default parameters.

**RNA sequencing and quantification of gene expression**. Quality of reads was assessed for each sample using FastQC. Fastq files were aligned to the reference Human genome hg38 with STAR1[59]. Reads mapping to multiple locations were removed. Gene expression was quantified using the full Gencode v26 annotation. STAR was used to obtain the number of reads associated to each gene in the Gencode v26 database (restricted to protein-coding genes, antisense and lincRNAs). Bioconductor DESeq package[60] was used to import raw HTSeq counts for each sample into R statistical software and extract the count matrix. After normalizing for library size, the count matrix was normalized by the coding length of genes to compute FPKM scores (number of fragments per kilobase of exon model and millions of mapped reads). Bigwig visualization files were generated using STAR and the bam2wig python script[61].

**Unsupervised analysis**. Bioconductor edgeR package[62] was used to import raw HTSeq counts into R statistical software, and compute normalized log2 CPM (counts per millions of mapped reads) using the TMM, weighted trimmed mean of *M*-values, as normalization procedure. The normalized expression matrix from the 1000 most variant genes (based on standard deviation) or from a custom 250 gene list was used to classify the samples according to their gene expression patterns using PCA, hierarchical clustering and consensus clustering. Standard R functions were used to perform the PCA and hierarchical clustering (with Euclidean distance and Ward method). Consensus clustering (Bioconductor ConsensusClusterPlus package) was used to examine the stability of the clusters. Consensus partitions of the data set were established in *K* clusters (for *K* = 2, 3, …, 8), on the basis of 1000 resampling iterations (80% of genes, 80% of sample) of hierarchical clustering, with Pearson's dissimilarity as the distance metric and Ward's method for linkage analysis. The cumulative distribution functions (CDFs) of the consensus matrices were used to determine the optimal number of clusters (*K* = 3 for instance), considering both the shape of the functions and the area under the CDF curves.

**Differential expression analysis**. The Bioconductor limma package[63] was used to test for differential expression using the voom transformation. Only genes expressed in at least one sample (FPKM ≥ 0.1) were tested to improve the statistical power of the analysis. A *q* value threshold of ≤0.1 and a minimum log2 fold change of 1 were used to define differentially expressed genes.

**Pathway enrichment analysis**. Hypergeometric tests were used to identify gene sets from the custom gene list, overrepresented of up- or downregulated genes, correcting for multiple testing with the Benjamini–Hochberg procedure.

**Reporting summary**. Further information on research design is available in the Nature Research Reporting Summary linked to this article.

## Data availability

The WES and RNA sequencing data have been deposited in the EGA database under the accession code EGAS00001004272. The WES and RNA sequencing data referenced during the study are available in a public repository using the https://www.ebi.ac.uk/ega/studies/EGAS00001004272 hyperlink. The source data underlying Figs. 2, 3, 4, 5, 6 and Supplementary Figs 3, 5, 6, 7 are provided at https://www.ebi.ac.uk/ega/studies/EGAS00001004272. All the other data supporting the findings of this study are available within the article and its supplementary information files and from the corresponding authors upon reasonable request. A reporting summary for this article is available as a Supplementary Information file.

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

## Acknowledgements

We thank Dr. Patrycja Pawlikowska and Tala Tayoun for very helpful discussions and critical reading of the manuscript and Integragen GeCo (Evry, France) and Dr. Marc Deloger for bioinformatics analysis. We are thankful to the members of the CTCTrap consortium. We are very grateful to the patients and their families. The study was supported by ANR-15-CE17-0006-01, CTCTrap FP7 HEALTH #305341, and the Fondation ARC pour la Recherche sur le Cancer (no. PJA20171206530). V.F. and E.P. were supported by the Fondation pour la Recherche Médicale (no. FDT20160435543 FDT20150532072).

## Author contributions

V.F. conducted experiments and contributed to experimental design, data analysis, and manuscript editing with the assistance of E.P., M.O., O.D., L.B.-S and C.H. V.M. and J.-Y.S. performed immunohistopathology experiments and analysis. K.A., D.T., M.N.C., C.N., V.L., K.F., and Y.L. supported patient recruitment and sample management, and provided clinical support for the study. K.C.A., N.H.S., L.W.M.M.T., and N.M. contributed to the experimental design of diagnostic leukapheresis. O.D., L.B.-S., S.C., and J.-G.J. conducted in vivo experiments. F.F. directed the research and edited the manuscript. V.F., F.F., Y.L., and J.-G.J. contributed to data analysis.

## Competing interests

The authors declare no competing interests.
