## [Peer Review File · Nature Communications]

Reviewers' comments:

Reviewer #1 (Expertise: Prostate cancer, models, Remarks to the Author):

Therapy-induced neuroendocrine transdifferentiation (NED) is currently a major clinical challenge in prostate cancer treatment and the experimental models to address this lethal problem are limited. The present authors collected and enriched circulating tumor cells (CTC) from an advanced castrate resistant prostate cancer (CRPC) patient with Gleason score 9 and successfully engrafted a SQ tumor (CDX), followed by establishing a CDX-derived cell line. The neuroendocrine phenotype of both CDX and CDX-derived cell line were further confirmed by histological and genetic characterization. These CDX models may provide a unique tool for effective drug screening in NED management.

Despite the significant strengths of the paper, several issues arise that need to be addressed:

1. Previous work has shown that NED induced by androgen depletion is a reversible process. The CDX in the present study was derived from a post-ADT CRPC patient specimen. Can the CDX-derived cell lines' phenotype be reversed by culturing cells in normal serum-containing medium?
2. NED includes two major cell types: Focal type refers to neuroendocrine foci found in prostate adenocarcinoma and Universal type is a small cell neuroendocrine carcinoma. CD44 is 100% positive in prostate small cell NE carcinoma but was detectable only in a minority of small cell NE carcinoma from other organs, therefore CD44 can serve as a useful biomarker to distinguish the origin of prostate small cell NE carcinoma. Do CDX and CDX-derived cells express CD44?
3. If CDX is a true CRPC-NED, the tumors should grow under castration condition. The authors should clarify the usage/need of a 10 mg testosterone capsule in establishing CDX tumor growth in vivo.
4. CDX-derived cells are resistant to both docetaxel and enzalutamide treatments in cultures. Is CDX tumor growth also resistant to docetaxel or enzalutamide treatment in vivo?
5. The authors described the use of tumor fragments for serial passage of CDX in the Methods section, but there were no data shown in the Results section.
6. What are the tumor take rates by using CDX tumor fragments and by CDX-derived cell lines?
7. Certain common signaling pathways have been shown to be activated in NED, they include PKA/CREB, STAT3, MAPKs, PI3K-Akt-mTOR, Wnt and other pathways. Do any of these NED signaling pathways show up in CDX and CDX-derived cells?
8. Several molecular components are also involved in driving NED, besides TMPRSS2-ERG fusion and the loss of TP53, TPEN, and RB1. Did the authors find any gene amplification for AURKA, MYCN, TIMP-1, survivin and Bcl-2 or suppression for REST in CDX? This will be important to assess and describe.

9. PC3 is characterized as a small cell neuroendocrine carcinoma cell line, while LuCaP49 is a neuroendocrine PDX derived from omental fat metastasis. These should be discussed and compared to CDX and CDX-derived cells in the study.

10. Base on the literature below, tumor volume should be calculated using formula $V=0.5 \times \text{length} \times \text{width} \times \text{width}$.

Reference: Lab Anim (NY). 2013 Jun;42(6):217-24. Estimation of rat mammary tumor volume using caliper and ultrasonography measurements. Faustino-Rocha A, Oliveira PA, Pinho-Oliveira J, Teixeira-Guedes C, Soares-Maia R, da Costa RG, Colaço B, Pires MJ, Colaço J, Ferreira R, Ginja M.

Reviewer #2 (Expertise: CTCs, Remarks to the Author):

Faugeroux et al. describe a CTC derived eXplant (CDX) model from castrate resistant prostate cancer (CRPC) using leukapheresis. This CDX is analyzed by genomic analysis and compared to primary tumor (PT) and CTCs to understand AR resistance. 7 patients with leukapheresis were processed yielding 1 CDX. Overall, the authors report the generation of a prostate CDX but the genomic analysis shows so many differences between primary tumor samples, CDX, and CTCs it is really hard to make any clear conclusions with a N=1.

Comments

1) The abstract is very hard to understand and significant work should be spent on improving clarity. Is the point to say that the CDX only 4% of PT and 0.3% CTC mutations were seen in the CDX while CNAs were more commonly seen? Given this huge differences how does this provide insight on the "sequence" of these features?

2) In the introduction, the authors provide their very strong opinions on different prostate cancer models and suggest that GEMs are not useful relative to PDX models, and in general, that all models are insufficient. I would recommend softening the tone of these opinions and focus on what a CDX model provides in addition to these other models.

3) CTC cell line generation is reported as being very limited, but there is one colon CTC cell line reported (reference 24) and 6 breast CTC cell lines (reference 25) reported. The authors report the generation of a single prostate CDX here so I'm not sure they have really overcome these difficulties

with their strategy. Again the tone of this paragraph is very negative on the work that has been published before.

4) CDX had low PSA and AR compared to PT biopsies indicating tumor evolution, however, the authors then describe 10% of the PT was neuroendocrine. "evolution" may not be the most appropriate term, but this would suggest selection based on what they report in the PT

5) The authors report an extraordinary level of mutational heterogeneity in the primary tumor with 87% unique to a PT biopsy. This seems extraordinarily high and I am concerned that the variant caller may be providing too many false positives. Recommend performing other combination of variant caller with aligner (See Alioto TS et al. Nature Comm 2015)

6) Fig 7 clearly CDX cell line is resistant to enzalutamide and maybe slightly more resistant to docetaxel, but is it significant?

Reviewer #3 (Expertise: CRPC-NE, clinical, Remarks to the Author):

The study explore the use of CTC to derive PDXs from patients with advanced prostate cancer. This is an important approach to help expand the limited models available to prostate cancer researchers. Unfortunately, we do not learn much about how to extend this approach or what we now know should or should not work.

This study demonstrates the difficulty of establishing CTXs with success in 1 out of around 22 patients with CRPC. The case that worked shows an interesting transdifferentiation pattern going from adenocarcinoma with some neuroendocrine features to a full blown neuroendocrine tumor. Mutation detection was performed in this one example and shows maintainance of truncal lesions (e.g., NKX3.1 and TMRPSS2-ERG). Interestingly only one of the primary tumor samples harbored losses at TP53, which was seen to be a common feature of the CTX supporting this as one of the selected drivers. Other common losses in the CTXs included PTEN and RB1. Evolutionary analysis suggests that one primary tumor sample was a key driver of the metastatic event leading to the CTX.

The cBioportal analysis is not useful as this is aggregated. It would be more useful to look at this in the setting of specific studies referring to the specific cohorts. Do they formally compare for example the Robinson (Cell) data and the TCGA?

It is good to see that the CTX was resistant to ARSi. It would be interesting to know if they were sensitive to PARPi or Platinum.

Figure 1 shows multiple phases of the CTX. It is interesting to note that the cell that is described to act like a small cell-neuroendocrine cancer is not histologically small cell. In fact it has too much cytoplasm. Performing RNAseq on this data set would be highly valuable to monitor AR and NE signaling.

The text needed to be carefully edited for clarity. For example, results in the abstract are not clear to readers. The introduction reads more like a book chapter (e.g., section of CTC). It would be most helpful to focus on the study at hand. There are numerous spelling errors (e.g. "the remainder of the tumor was used for Alu sequence detection...")

LNCAP is not considered a model of CRPC as suggested in the introduction.

Reviewer #4 (Expertise: WES, Remarks to the Author):

The genomics and bioinformatics methods employed in this study are appropriate. However, as detailed below, there are several major technical concerns related to tumor genotyping:

Major comments:

1. The authors applied lenient filtration settings for identifying somatic variants: 5% VAF threshold with at least 3x mutant reads in the tumor; <4% VAF threshold with <3 mutant reads in the paired normal. Requiring only 3 reads to support a variant is within the realm of technical noise on Illumina instruments without the use of molecular barcoding and/or a more stringent PHRED quality score threshold (e.g., the authors used only Q20 [1% error rate]). Given the relatively low sequencing depths achieved in this study, the use of more stringent thresholds in the tumor and paired normal would increase confidence in the somatic variants identified (e.g., \geq Q30, a threshold of 5 or even greater than 5 mutant reads in the tumor; 0% VAF in the normal).

2. The authors apply a strange and seemingly arbitrary "in-house" somatic mutation score. The need for this score is not established, and the justification for the score is not discussed. On what

statistical basis should the VAF be multiplied by the number of mutant reads? The authors should either justify this score with empirical data (or even better, with a rigorous statistical argument), or remove it.

3. The authors should analyze and discuss the distribution of VAFs identified. This will help to understand how the number and distribution of variants change as a function of different VAF cutoffs. For example, if most of the identified variants are in a long tail of lower VAF events, many of them may be false positives.

4. The authors identify significant heterogeneity among (1) diagnostic biopsies (Fig. 2A), (2) tumor biopsies, CTCs, CDX and the cell line (Fig. 2B), and (3) CTCs (Fig. 3A), all of which are derived from the same patient. If true, this is an important biological result. However, the high amount of variability may also be attributed to, at least in part, the issues raised above. Targeted re-sequencing at much higher sequencing depths would be a good way to validate the authors' results and address some of the above concerns.

5. The authors attempt to estimate the FPR of variant allele detection ("Allele drop-out and false positive rate estimation"). As shown in Supplementary Fig. 4A, by applying the same analytical workflow and filtration settings used to define tumor variants, they estimate there to be 93 SNVs and 52 indels in their bulk WBC (normal) sample (assuming I understand this analysis correctly). This is an extremely troubling number of FPs, especially seeing as (1) the physical coverage of the bulk WBC exome is only ~half of the coverage of tumor biopsy samples at 25x, and (2) this FP variant rate suggests that almost half of the median number of SNVs+indels identified in the tumor biopsies are FPs (and this is a lower bound given the preceding point).

6. The authors should provide a supplementary table containing all identified variants, including the following data: genomic position, nucleotide change, codon number, amino acid change, type of variant, VAF, number of mutant reads in the tumor, number of total reads in the tumor, number of mutant reads in the normal, number of total reads in the normal.

Minor comments:

1. Supplementary Fig. 4: Please provide the median depth in panel A; the mean depth is skewed by outliers and is often misleading.

2. The authors' use of "Constitutional DNA" is unclear and unconventional.

3. The FPR for variant detection noted in the Results is 4.4×10^6 . This appears to be a typo (4.4×10^{-6}).

Title: "Genetic characterization of a Unique Neuroendocrine Transdifferentiation Prostate Circulating Tumor Cell - Derived eXplant (CDX) Model".

Response to Reviewers

Reviewer #1:

Therapy-induced neuroendocrine transdifferentiation (NED) is currently a major clinical challenge in prostate cancer treatment and the experimental models to address this lethal problem are limited. The present authors collected and enriched circulating tumor cells (CTC) from an advanced castrate resistant prostate cancer (CRPC) patient with Gleason score 9 and successfully engrafted a SQ tumor (CDX), followed by establishing a CDX-derived cell line. The neuroendocrine phenotype of both CDX and CDX-derived cell line were further confirmed by histological and genetic characterization. These CDX models may provide a unique tool for effective drug screening in NED management.

Despite the significant strengths of the paper, several issues arise that need to be addressed:

1. Previous work has shown that NED induced by androgen depletion is a reversible process. The CDX in the present study was derived from a post-ADT CRPC patient specimen. Can the CDX-derived cell lines' phenotype be reversed by culturing cells in normal serum-containing medium?

The CDX-derived cell line has been established and is cultured in normal serum-containing medium. No reversion of the phenotype is observed.

The culture medium (Advanced DMEM/F12 medium supplemented with 10% FBS, 1% antibiotics and 1% ultraglutamine) used for the establishment of the CDX-derived cell line was described in the Patients and Methods section page 17 line 14-17. The same normal serum-containing medium is used for culturing the permanent CDX-derived cell line. This point is now clarified in the Patients and Methods section page 17 lines 20-21 and in the Result section page 25 line 20.

2. NED includes two major cell types: Focal type refers to neuroendocrine foci found in prostate adenocarcinoma and Universal type is a small cell neuroendocrine carcinoma. CD44 is 100% positive in prostate small cell NE carcinoma but was detectable only in a minority of small cell NE carcinoma from other organs, therefore CD44 can serve as a

useful biomarker to distinguish the origin of prostate small cell NE carcinoma. Do CDX and CDX-derived cells express CD44?

CD44 expression has been examined. CD44 is strongly expressed in the CDX and the CDX-derived cell line. This result is now presented in the Results section page 25 line 19 and in the Legend section for Figure 1 page 43 line 7. Images of CD44 staining of PT specimens, the CDX and the CDX-derived cell line are now shown in Figure 1.

3. If CDX is a true CRPC-NED, the tumors should grow under castration condition. The authors should clarify the usage/need of a 10 mg testosterone capsule in establishing CDX tumor growth *in vivo*.

The CDX is growing under castration condition as shown in the *in vivo* experiment now presented in Figure 8. There was no significant difference in tumor growth between castrated and non-castrated animals.

A 10 mg testosterone capsule was used in the first mouse implanted with CTCs and in successive generations of mice until passage 7. At passage 7, testosterone dependence was tested by comparing tumor growth in groups of animals supplemented or not with testosterone. Tumor growth was similar in presence or absence of testosterone. This information is now included in the Patients and Methods section page 10 lines 13-15 and the Results section page 25 lines 7-10.

4. CDX-derived cells are resistant to both docetaxel and enzalutamide treatments in cultures. Is CDX tumor growth also resistant to docetaxel or enzalutamide treatment *in vivo*?

In vivo experiments have been performed. CDX tumor growth is also resistant to docetaxel and enzalutamide treatments. No significant difference in tumor growth was observed between treated and control tumors. Additionally, as requested by Reviewer 3 we show that the CDX is resistant to olaparib, which could also be predicted by the absence of bi-allelic alteration of DNA repair pathways genes. These results are now included in the Results section page 33 lines 12-18. Figure 8 presenting *in vivo* experiment results (legend for Figure 8 page 47 line 15-21) and the *in vivo* drug assay method (Patient and Method section page 18 lines 4-23 and page 19 lines 1-3) are now included.

5. The authors described the used of tumor fragments for serial passage of CDX in the Methods section, but there were no data shown in the Results section.

The use of tumor fragments for serial passages is now mentioned in the Results section page 25 lines 5-6.

6. What are the tumor take rates by using CDX tumor fragments and by CDX-derived cell lines?

Tumor fragments were used for the first passages of the CDX. When the CDX-derived cell line was established, we compared the take rate, latency and size of CDX tumors by using tumor fragments and the CDX-derived cell line. Results are presented below and show the higher efficiency of tumor fragments compared to CDX-derived cell line.

This point is now mentioned in the Patients and Methods section page 10 lines 15-18.

	Latency tumors for 60-200 mm ³	% tumors (60-200 mm ³)	Latency tumors for >500mm ³	% tumors >500 mm ³	Take rate (% growing tumors)
CDX-derived cell line	53 days	50%	90 days	50%	50%
CDX tumor fragments	40 days	50%	68 days	60%	100%

7. Certain common signaling pathways have been shown to be activated in NED, they include PKA/CREB, STAT3, MAPKs, PI3K-Akt-mTOR, Wnt and other pathways. Do any of these NED signaling pathways show up in CDX and CDX-derived cells?

To address this comment, we performed RNA sequencing of the prostate adenocarcinoma LNCaP cell line, the CDX and the CDX-derived cell line. Results are presented in the Results section page 26 lines 9-23 and page 27 lines 1-9. Unsupervised hierarchical clustering of the 1000 most variant genes evidenced two clusters, the first composed of LNCaP samples and the second of the CDX and the CDX-derived cell line (shown in Supplementary Figure 3). Then we focused on 250 functional genes relevant for CRPC-NE progression according to Beltran et al, Nat Med 2016 and Aggarwal et al, J Clin Oncol 2018. These data further confirmed the two clusters, and the similarity of the transcriptional profiles of the CDX and the cell line (shown in Fig 2A). By supervised analysis of genes differentially expressed between LNCaP cells and the CDX, genes involved in NED signaling pathways including E2F transcription factors and Wnt were significantly upregulated while AR and Notch pathways were downregulated (Fig 2B). As shown in Figure 2A, some genes of the CREB pathway were overexpressed but upregulation of the whole pathway was not statistically significant. Similarly, the supervised analysis of genes differentially expressed between LNCaP cells and the CDX do not show an upregulation of MAPKs, PI3K-Akt-mTOR pathways. Genes implicated in neural development and *CHGA* and *SYP* genes were overexpressed (Fig 2C). Transcriptional regulators including *STAT3*, *ASCL1*, *SOX2*,

POU3F2, *FOXA2*, *FOXA1*, *PDX1* and *REST* genes were deregulated (Fig 2C). *EZH2* and *TIMP-1* genes were strongly overexpressed while *PTEN*, *TP53*, *RB1* and *CYLD* tumor suppressor genes were under-expressed. Overall, the transcriptional profiling indicated that numerous genes and signaling pathways that are involved in driving NED are deregulated in the CDX and CDX-derived cell line.

-Beltran H, Prandi D, Mosquera JM, Benelli M, Puca L, Cyrta J, Marotz C, Giannopoulou E, Chakravarthi BV, Varambally S, Tomlins SA, Nanus DM, Tagawa ST, Van Allen EM, Elemento O, Sboner A, Garraway LA, Rubin MA, Demichelis F. Divergent clonal evolution of castration-resistant neuroendocrine prostate cancer. *Nat Med*. 2016 Mar;22(3):298-305.

-Aggarwal R, Huang J, Alumkal JJ, Zhang L, Feng FY, Thomas GV, Weinstein AS, Friedl V, Zhang C, Witte ON, Lloyd P, Gleave M, Evans CP, Youngren J, Beer TM, Rettig M, Wong CK, True L, Foye A, Playdle D, Ryan CJ, Lara P, Chi KN, Uzunangelov V, Sokolov A, Newton Y, Beltran H, Demichelis F, Rubin MA, Stuart JM, Small EJ. Clinical and Genomic Characterization of Treatment-Emergent Small-Cell Neuroendocrine Prostate Cancer: A Multi-institutional Prospective Study. *J Clin Oncol*. 2018 Aug 20;36(24):2492-2503.

8. Several molecular components are also involved in driving NED, besides *TMPRSS2-ERG* fusion and the loss of *TP53*, *TPEN*, and *RB1*. Did the authors find any gene amplification for *AURKA*, *MYCN*, *TIMP-1*, *survivin* and *Bcl-2* or suppression for *REST* in CDX? This will be important to assess and describe.

Studies focusing on NEPC have reported loss of *REST*, amplification of *AURKA*, *MYCN*, *TIMP-1*, *survivin* or *Bcl-2*, as potential additional drivers of NEPC. The presence of deleterious mutations and/or copy number loss in DNA repair pathway genes (*BRCA1*, *BRCA2*, *ATM*, *CDK12*, *RAD51*, *PALB2*, *FANCA*, *CHEK2*, *MLH1*, *MSH2*, *MLH3*, and *MSH6*) was also found almost mutually exclusive with treatment-emergent small cell neuroendocrine tumors (Aggarwal *et al*, *J Clin Oncol* 2018).

To address this question, we examined genomic data and did not find *MYCN*, *AURKA*, *TIMP-1* or *survivin* gene amplification. However, loss of *REST* was found in the CDX, the CDX-derived cell line and CTCs. No biallelic loss of function alteration of DNA repair genes was observed, which was consistent with the study by Aggarwal *et al*. As suggested by the reviewer, we have now included these data in the Results section page 31 line 20 and in the Discussion section page 39 lines 9-14. By transcriptional profiling, no overexpression of *AURKA*, *MYCN*, *survivin* and *Bcl-2* genes was found while *TIMP-1* was significantly up-regulated (qvalue = 0.0008) as shown in the new Figure 2C.

9. PC3 is characterized as a small cell neuroendocrine carcinoma cell line, while LuCaP49 is a neuroendocrine PDX derived from omental fat metastasis. These should be discussed and compared to CDX and CDX-derived cells in the study.

We have now discussed and compared the PC3 cell line and LuCaP49 PDX to our CDX/CDX-derived cell line model. This is included in the Discussion section page 36 lines 11-18.

10. Base on the literature below, tumor volume should be calculated using formula $V=0.5 \times \text{length} \times \text{width} \times \text{width}$.

We are sorry for this typo. Tumor volume was indeed calculated using this formula (tumor length x tumor width²) /2 not (tumor length x tumor width)²/2 formula.

We have now rectified this error in the Patients and Methods section page 10 line 9.

Reference: Lab Anim (NY). 2013 Jun;42(6):217-24. Estimation of rat mammary tumor volume using caliper and ultrasonography measurements. Faustino-Rocha A, Oliveira PA, Pinho-Oliveira J, Teixeira-Guedes C, Soares-Maia R, da Costa RG, Colaço B, Pires MJ, Colaço J, Ferreira R, Ginja M.

Note: Owing to the application of more stringent bioinformatics settings recommended by Reviewer 2 and Reviewer 4, a large number of variants with a low frequency have been lost. We have therefore removed the Figures 2D and 3B and replaced the Figure 2D by the supplementary Figure 6.

Reviewer #2:

Faugeroux *et al.* describe a CTC derived eXplant (CDX) model from castrate resistant prostate cancer (CRPC) using leukapheresis. This CDX is analyzed by genomic analysis and compared to primary tumor (PT) and CTCs to understand AR resistance. 7 patients with leukapheresis were processed yielding 1 CDX. Overall, the authors report the generation of a prostate CDX but the genomic analysis shows so many differences between primary tumor samples, CDX, and CTCs it is really hard to make any clear conclusions with a N=1.

Comments

1) The abstract is very hard to understand and significant work should be spent on improving clarity. Is the point to say that the CDX only 4% of PT and 0.3% CTC mutations were seen in the CDX while CNAs were more commonly seen? Given this huge differences how does this provide insight on the “sequence” of these features?

We have modified the abstract and improved its clarity (page 3).

In response to Reviewer's comment n°5 and to Reviewer 4 we have now used more stringent post-filtering steps for somatic variants identification. Using these new filtration settings, we find that 16% of PT and 1.9% CTC mutations are present in the CDX and the derived-cell line. In contrast, 83% of PT CNAs are conserved in the CDX and the cell line. One aim of the study is to examine the genetic basis of the tumorigenic activity of prostate CTCs in this model. These data show that in spite of the mutational diversity of PT and CTCs, only a small fraction of the PT and CTC mutations are associated with the tumorigenic activity of CTCs. In contrast, PT specimens harbored a limited number (n=6) of CNAs but almost all are associated with the tumorigenicity of CTCs.

These data provide information on the proportion and nature of genetic events in the primary tumor that are associated with the tumorigenicity of CTCs. However, we recognize that they do not provide information on the sequence of events. We are also fully aware that this data is representative of only one model and cannot be generalized.

Phylogenetic tree reconstruction informs the sequence of events and identifies drivers of the neuroendocrine transdifferentiation of CRPC. While *TMPRSS2-ERG* fusion was clonal, we show that the three genetic events i.e. loss of *PTEN*, *RB1* and *TP53*, which are hallmarks of neuroendocrine transdifferentiation, were typically acquired in CTCs and conserved in the CDX/cell line model. Loss of *PTEN* and *RB1* was exclusively acquired in CTCs and the CDX/cell line. In contrast, *TP53* loss was detected in five of the eight primary tumor specimens and in two different configurations on chromosome 17 (i.e loss of 17p12-tel and 17p12-13). Interestingly, only loss of the 17p12-tel segment of *TP53* was conserved in CTCs, the CDX and the cell line. This infers that loss of *TP53* (17p12-tel segment) preceded loss of *PTEN* and *RB1* in this model.

2) In the introduction, the authors provide their very strong opinions on different prostate cancer models and suggest that GEMs are not useful relative to PDX models, and in general, that all models are insufficient. I would recommend softening the tone of these opinions and focus on what a CDX model provides in addition to these other models.

We fully understand this comment. In our introduction, we did not want to criticize current models that have actually led to important progress but only to present the needs and the difficulty of establishing CRPC models. We have now changed the tone used to introduce existing models (introduction section page 5 lines 7-20) and presented the specific contribution of CDX models (page 6 lines 21-23 and page 7 lines 1-3).

3) CTC cell line generation is reported as being very limited, but there is one colon CTC cell line reported (reference 24) and 6 breast CTC cell lines (reference 25) reported. The authors report the generation of a single prostate CDX here so I'm not sure they have really overcome these difficulties with their strategy. Again the tone of this paragraph is very negative on the work that has been published before.

We have now modified the sentence used to introduce CTC cell lines (introduction page 7 lines 8-12) and changed the tone used to present the work that has been published before. We did not want to criticize the work done previously, but only present the difficulty of establishing these models. The approach of using a DLA has made the establishment of the model described here possible. But we are fully aware that the generation of a single model does not allow us to conclude that the use of DLAs can help circumvent all difficulties related to CTCs.

This is now mentioned in the Discussion section page 36 line 23 and page 37 line 1-2.

4) CDX had low PSA and AR compared to PT biopsies indicating tumor evolution, however, the authors then describe 10% of the PT was neuroendocrine. "evolution" may not be the most appropriate term, but this would suggest selection based on what they report in the PT. The word "evolution" has been removed (Result section page 25 line 15).

5) The authors report an extraordinary level of mutational heterogeneity in the primary tumor with 87% unique to a PT biopsy. This seems extraordinarily high and I am concerned that the variant caller may be providing too many false positives. Recommend performing other combination of variant caller with aligner (See Alioto TS et al. Nature Comm 2015).

To address this comment and that of Reviewer 4 we have now used more stringent filtration settings for identifying somatic variants (e.g., Qhred ≥ 30 , a threshold of 5 mutant reads in tumor samples; 0% VAF in the normal). The identification of somatic variants has been completely re-done with these new thresholds. New data are now presented in the Abstract, Patients and Methods section page 14 lines 7-9, Results section pages 27-33, Figures 3, 4, 5, 6 and Supplementary Figure 5.

Owing to the application of more stringent filters, a large number of variants with a low VAF have been lost. We have therefore removed the Figures 2D and 3B and replaced the Figure 2D by the supplementary Figure 6.

6) Fig 7 clearly CDX cell line is resistant to enzalutamide and maybe slightly more resistant to docetaxel, but is it significant?

Like PC-3, the CDX-derived cell line is highly resistant to enzalutamide since the IC50 is never reached in comparison to LNCaP, which is sensitive. The CDX-derived cell line is also resistant to docetaxel with an IC50 that is significantly higher than that of PC-3 and LNCaP. Statistics are now shown in an additional Figure 9B.

Reviewer #3:

The study explores the use of CTC to derive PDXs from patients with advanced prostate cancer. This is an important approach to help expand the limited models available to prostate cancer researchers. Unfortunately, we do not learn much about how to extend this approach or what we now know should or should not work.

This study demonstrates the difficulty of establishing CTXs with success in 1 out of around 22 patients with CRPC. The case that worked shows an interesting transdifferentiation pattern going from adenocarcinoma with some neuroendocrine features to a full blown neuroendocrine tumor. Mutation detection was performed in this one example and shows maintenance of truncal lesions (e.g., NKX3.1 and TMRPSS2-ERG). Interestingly only one of the primary tumor samples harbored losses at TP53, which was seen to be a common feature of the CTX supporting this as one of the selected drivers. Other common losses in the CTXs included PTEN and RB1. Evolutionary analysis suggests that one primary tumor sample was a key driver of the metastatic event leading to the CTX.

The cBioportal analysis is not useful as this is aggregated. It would be more useful to look at this in the setting of specific studies referring to the specific cohorts. Do they formally compare for example the Robinson (Cell) data and the TCGA?

We have now focused the cBioportal analysis on eight specific studies (instead of 14) that reported CRPC and prostate neuroendocrine carcinoma cohorts (listed below). We have compared the gene alteration frequencies found in our study to those published in the eight studies. The results of this new analysis are presented in the Results section page 32 lines 5-23, page 33 lines 1-7 and in the Figures 7A, B, C, D and supplementary Figures 8, 9 and 10.

It is noteworthy that we do not re-analyze the data reported in these studies with our tools to perform this comparison.

-Beltran H, Prandi D, Mosquera JM et al. Divergent clonal evolution of castration-resistant neuroendocrine prostate cancer. Nat Med 2016; 22: 298-305.

-Grasso CS, Wu YM, Robinson DR et al. The mutational landscape of lethal castration-resistant prostate cancer. *Nature* 2012; 487: 239-243.

-Baca SC, Prandi D, Lawrence MS et al. Punctuated evolution of prostate cancer genomes. *Cell* 2013; 153: 666-677.

-Robinson D, Van Allen EM, Wu YM et al. Integrative clinical genomics of advanced prostate cancer. *Cell* 2015; 161: 1215-1228.

-Kumar A, Coleman I, Morrissey C et al. Substantial interindividual and limited intraindividual genomic diversity among tumors from men with metastatic prostate cancer. *Nat Med* 2016; 22: 369-378.

-Abida W, Armenia J, Gopalan A et al. Prospective Genomic Profiling of Prostate Cancer Across Disease States Reveals Germline and Somatic Alterations That May Affect Clinical Decision Making. *JCO Precis Oncol* 2017; 2017.

-Armenia J, Wankowicz SAM, Liu D et al. The long tail of oncogenic drivers in prostate cancer. *Nat Genet* 2018; 50: 645-651.

-Abida W, Cyrta J, Heller G et al. Genomic correlates of clinical outcome in advanced prostate cancer. *Proc Natl Acad Sci U S A* 2019; 116: 11428-11436.

It is good to see that the CTX was resistant to ARSi. It would be interesting to know if they were sensitive to PARPi or Platinum.

In vivo experiments have been performed to evaluate whether the CDX is sensitive to olaparib. This result is presented in the Result section page 33 lines 12-18. As shown in Figure 8, no significant difference in the CDX tumor growth of olaparib-treated as compared to control tumors was observed. This result shows that, as predicted by the absence of predictive factors of sensitivity (ie bi-allelic alteration of DNA repair pathways genes), the CDX is resistant to olaparib.

Figure 1 shows multiple phases of the CTX. It is interesting to note that the cell that is described to act like a small cell-neuroendocrine cancer is not histologically small cell. In fact it have too much cytoplasm. Performing RNAseq on this data set would be highly valuable to monitor AR and NE signaling.

To address this comment, we performed RNA sequencing of the prostate adenocarcinoma LNCaP cell line, the CDX and the CDX-derived cell line. Results are presented in the Results section page 26 lines 9-23 and page 27 lines 1-9. Unsupervised hierarchical clustering of the 1000 most variant genes evidenced two clusters, the first composed of LNCaP samples and the second of the CDX and the derived cell line (shown in Supplementary Figure 3). We focused on 250 functional genes relevant for CRPC-NE progression according to Beltran et al, *Nat Med* 2016 and Aggarwal et al, *J Clin Oncol* 2018. These data further confirmed the two clusters, and the similarity of the transcriptional profiles of the CDX and the derived cell line (shown in Figure 2A). By supervised analysis of genes differentially expressed between LNCaP cells and the CDX, genes involved in NED signaling pathways including E2F transcription factors and Wnt were significantly up-regulated while AR and Notch pathways

were downregulated (Figure 2B). As shown in Figure 2A, some of *CREB* genes were overexpressed but upregulation of the whole pathway was not statistically significant. Similarly, the supervised analysis of genes differentially expressed between LNCAP cells and the CDX do not show an upregulation of MAPKs and PI3K-Akt-mTOR pathways. Genes implicated in neural development and *CHGA* and *SYP* genes were overexpressed (Figure 2C). Transcriptional regulators including *STAT3*, *ASCL1*, *SOX2*, *POU3F2*, *FOXA2*, *FOXA1*, *PDX1* and *REST* genes were deregulated (Figure 2C). *EZH2* and *TIMP-1* genes were strongly overexpressed while *PTEN*, *TP53*, *RB1* and *CYLD* tumor suppressor genes were under-expressed. Overall, the transcriptional profiling indicated that AR signaling is downregulated while numerous genes and signaling pathways involved in driving NED are deregulated in the CDX and CDX derived cell line.

The text needed to be carefully edited for clarity. For example, results in the abstract are not clear to readers. The introduction reads more like a book chapter (e.g., section of CTC). It would be most helpful to focus on the study at hand. There are numerous spelling errors (e.g. “the remainder of the tumor was used for Alu sequence detection...”).

We did our best to improve the clarity of the abstract as well as the text throughout the manuscript. We have simplified the introduction to focus on the study and corrected spelling errors.

LNCAP is not considered a model of CRPC as suggested in the introduction.

We recognize our mistake. This is now removed from the introduction.

Note: Owing to the application of more stringent bioinformatics settings recommended by Reviewers 2 and 4, a large number of variants with a low frequency have been lost. We have therefore removed the Figures 2D and 3B and replaced the Figure 2D by the supplementary Figure 6.

Reviewer #4:

The genomics and bioinformatics methods employed in this study are appropriate. However, as detailed below, there are several major technical concerns related to tumor genotyping:

Major

comments:

1. The authors applied lenient filtration settings for identifying somatic variants: 5% VAF

threshold with at least 3x mutant reads in the tumor; <4% VAF threshold with <3 mutant reads in the paired normal. Requiring only 3 reads to support a variant is within the realm of technical noise on Illumina instruments without the use of molecular barcoding and/or a more stringent PHRED quality score threshold (e.g., the authors used only Q20 [1% error rate]). Given the relatively low sequencing depths achieved in this study, the use of more stringent thresholds in the tumor and paired normal would increase confidence in the somatic variants identified (e.g., \geq Q30, a threshold of 5 or even greater than 5 mutant reads in the tumor; 0% VAF in the normal).

We have now used the recommended filters for identifying somatic variants:

- Qphred \geq 30 (instead of 20)
- \geq 5 mutant reads (instead of 3)
- VAF=0 in the normal (instead of <4%)
- removal of « in-house » somatic score

New filtration settings are now mentioned in the Patients and Methods section page 14 lines 5-9.

According to these new thresholds, all the data related to the genomic analysis of primary tumor biopsies, CTCs, CDX and CDX-derived cell line have been modified in the Results section pages 27 to 33. Figures 3 to 7 and Supplementary Figure 5 have also been modified.

2. The authors apply a strange and seemingly arbitrary “in-house” somatic mutation score. The need for this score is not established, and the justification for the score is not discussed. On what statistical basis should the VAF be multiplied by the number of mutant reads? The authors should either justify this score with empirical data (or even better, with a rigorous statistical argument), or remove it.

As recommended by the reviewer and stated above we have now removed the “in house” somatic mutation score (Patients and Methods section page 14 lines 11-13).

3. The authors should analyze and discuss the distribution of VAFs identified. This will help to understand how the number and distribution of variants change as a function of different VAF cutoffs. For example, if most of the identified variants are in a long tail of lower VAF events, many of them may be false positives.

We fully understand this comment. We have analyzed the distribution of VAFs using our initial filtration settings and the new filtration settings recommended by the reviewer. This actually convinced us of the interest of using the new filtration settings.

The distribution of VAFs using the new filtration settings recommended by the Reviewer is shown in the additional Figure 1 attached (entitled “Additional Figure 1 in response to Reviewer 4 comment 3”).

4. The authors identify significant heterogeneity among (1) diagnostic biopsies (Fig. 2A), (2) tumor biopsies, CTCs, CDX and the cell line (Fig. 2B), and (3) CTCs (Fig. 3A), all of which are derived from the same patient. If true, this is an important biological result. However, the high amount of variability may also be attributed to, at least in part, the issues raised above. Targeted re-sequencing at much higher sequencing depths would be a good way to validate the authors’ results and address some of the above concerns.

We have changed our filtration settings for identifying somatic variants and used the settings recommended by the Reviewer. Therefore, the number of mutations identified in tumor biopsies, CTCs, the CDX and the CDX-derived cell line has dropped dramatically.

Using the new filtration settings we have now identified 205 (instead of 1394) mutations in the eight tumor biopsies, 2267 (instead of 20,227) mutations in the seven CTC samples, 80 (instead of 112) mutations in the CDX and 85 (instead of 131) in the cell line.

These new data are now presented throughout the Results section pages 27-33. Figures 3A, 3B, 3C, 4A, 4B, 5A, 5B, 5C, 6 and 7 have been modified. Supplementary Figures 5, 6, 8, 9, 10 have been modified.

Owing to the use of more stringent settings and the drastic drop in the number of identified mutations -which is now even below the values reported in CRPC tumor biopsies (median SNV in metastatic biopsies of 41 (range: 2-729) (Beltran et al, Nat Med 2016)) -we do not perform targeted re-sequencing, which was suggested to validate mutations present at low frequency and confirm the high amount of tumor variability.

Given that a large number of variants with a low VAF have been lost, we have removed the Figures 2D and 3B and replaced the Figure 2D by the supplementary Figure 6.

Beltran H, Prandi D, Mosquera JM, Benelli M, Puca L, Cyrta J, Marotz C, Giannopoulou E, Chakravarthi BV, Varambally S, Tomlins SA, Nanus DM, Tagawa ST, Van Allen EM, Elemento O, Sboner A, Garraway LA, Rubin MA, Demichelis F. Divergent clonal evolution of castration-resistant neuroendocrine prostate cancer. *Nat Med.* 2016 Mar;22(3):298-305.

5. The authors attempt to estimate the FPR of variant allele detection (“Allele drop-out and false positive rate estimation”). As shown in Supplementary Fig. 4A, by applying the same analytical workflow and filtration settings used to define tumor variants, they estimate there to be 93 SNVs and 52 indels in their bulk WBC (normal) sample (assuming I understand this analysis correctly). This is an extremely troubling number of FPs, especially seeing as (1) the

physical coverage of the bulk WBC exome is only ~half of the coverage of tumor biopsy samples at 25x, and (2) this FP variant rate suggests that almost half of the median number of SNVs+indels identified in the tumor biopsies are FPs (and this is a lower bound given the preceding point).

We fully understand this comment. SNVs and indels values obtained using the new filtration settings are now presented in the Supplementary Figure 5A.

In spite of these new filtration settings, the number of SNVs and indels in the WBC bulk remains relatively high (15 SNVs and 26 indels). We therefore sought to understand where this could come from and reviewed the whole experimental process.

Although the quality controls performed after WGA were good (in agreement with the manufacturer's recommendations and the data by Polzer *et al*), the quality control carried out by analyzing the sizes of the DNA fragments after MseI digestion showed an average smaller size of fragments indicative of DNA degradation. These data are presented in the additional Figure 2 attached (entitled "Additional Figure 2 in response to Reviewer 4 comment 5"). We therefore hypothesize that the duration of cytopheresis (2h for this patient) and the various manipulations (centrifugations, volume reductions, transfer in bags and then tubes) have led to a fragility and degradation of WBC and ultimately to lower quality DNA .

Regarding the second point raised by the Reviewer ("this FP variant rate suggests that almost half of the median number of SNVs+indels identified in the tumor biopsies is FPs"...), we think that the numbers of SNVs+indels in the WBC bulk should rather be compared to the numbers found in the CTC samples which were rigorously treated by the same WGA and post-WGA purification processes (which is not the case for tumor biopsies).

This comment of the Reviewer also led us to re-analyze the results obtained with the CTC bulk. As shown in the Additional Figure 2, the CTC bulk also harbors DNA fragments of small size indicative of DNA degradation. We have no explanation for the poor quality of this sample that should have been removed from the study after analyzing the DNA fragment sizes. In addition, we observed that a very large part of the variants detected in the CTC bulk were lost after the application of the new filters. These observations together with low coverage indicate that this sample is not of good quality and we therefore found it preferable to eliminate the data of this bulk from the manuscript.

-Polzer B, Medoro G, Pasch S, Fontana F, Zorzino L, Pestka A, et al. Molecular profiling of single circulating tumor cells with diagnostic intention. *EMBO Mol Med.* 2014;6(11):1371-86.

6. The authors should provide a supplementary table containing all identified variants, including the following data: genomic position, nucleotide change, codon number, amino acid

change, type of variant, VAF, number of mutant reads in the tumor, number of total reads in the tumor, number of mutant reads in the normal, number of total reads in the normal.

A supplementary table 3 which includes all the data mentioned above is now included (Results section page 28 lines 14-15).

Minor comments :

1. Supplementary Fig. 4: Please provide the median depth in panel A; the mean depth is skewed by outliers and is often misleading.

The median depth is now included in supplementary Fig. 5A.

2. The authors' use of "Constitutional DNA" is unclear and unconventional. "Constitutional DNA" has been changed to "germline DNA" throughout the manuscript.

3. The FPR for variant detection noted in the Results is 4.4×10^6 . This appears to be a typo (4.4×10^{-6}). The typo is corrected. Given the new filtration settings the FPR is now 1.3×10^{-6} .

Additional Figure 1- In response to Reviewer 4 comment 3

Additional Figure 2 - In response to Reviewer 4 comment 5

CTC-1

CTC-2

CTC-3

CTC-4

CTC-5

CTC-6

Bulk CTC

Bulk WBC

Well	Sample ID	Range	ng/uL	% Total	nmole/L	Avg. Size	%CV
E1	CTC-1	50 bp to 1000 bp	35,7186	291,7	99,368	592	38,34
E2	CTC-2	50 bp to 1000 bp	45,2041	88,9	126,155	590	36,82
E3	CTC-3	50 bp to 1000 bp	25,9821	134,4	78,119	547	42,59
E4	CTC-4	50 bp to 1000 bp	27,9359	85,4	78,122	588	36,88
E5	CTC-5	50 bp to 1000 bp	40,0187	83	105,453	625	33,68
E6	CTC-6	50 bp to 1000 bp	50,3506	91,6	142,355	582	36,04
E7	Bulk CTC	50 bp to 1000 bp	49,6283	99,9	241,47	338	41,14
E8	Bulk WBC	50 bp to 1000 bp	22,0826	98,9	99,195	366	47,99

Reviewers' comments:

Reviewer #1 (Remarks to the Author):

The authors have added a significant amount of new data requested by this reviewer and have adequately addressed all previously raised concerns. The paper, models described and molecular insight are considered to be an important contribution to improved understanding of the development of neuroendocrine transdifferentiation in CRPC.

Reviewer #2 (Remarks to the Author):

Overall, the authors have improved the manuscript and from a technical standpoint I believe they have addressed my concerns. However, the main concern I have is that this is an N=1 report and generalizability about the insight of genomic evolution remains to be determined. Also, I find there is still a sense of overly ambitious claims. For example, the statement "Here, we report for the first time the feasibility of using DLA-increased CTC yield for CDX establishment." I don't think one can say a single CDX establishment can be sufficient to demonstrate "increased CTC yield for CDX establishment"

Reviewer #3 (Remarks to the Author):

This reviewer is satisfied with all responses by the authors. The manuscript is much improved.

Figure 2A could be improved for presentation. The gene names are too small to read as is. Perhaps they could either use the current 2A in the supplement and create a condensed version where only the labels of categories are presented. Regardless, as this is an important figure some additional thought should be given to presenting figure 2.

Reviewer #4 (Remarks to the Author):

In their revised manuscript, Faugeroux and colleagues have addressed serious issues with their tumor genotyping criteria, resulting in more credible results. Although the manuscript has improved, the substantial discrepancy between the large number of somatic variants per CTC (median of 338 SNVs per exome) and the more realistic number of somatic variants per primary tumor (median of 28 SNVs per exome) suggests that more rigorous criteria are needed for genotyping the latter, as detailed below.

Major comments:

1. In their rebuttal, the authors cite Beltran et al. Nat Med 2016 as a reference for the median number of exonic SNVs expected in metastatic tumor samples from patients with castration-resistant neuroendocrine prostate cancer (n = 41). It is therefore surprising that the authors are seemingly content with a nearly 10-fold higher number of SNVs in their CTC exomes. In fact, in a highly relevant prior study that is not cited by the authors (PMID 24752078), Lohr and colleagues show that WGA using MDA significantly inflates the false positive rate of calling SNVs from CTCs, whether as single CTCs (~25/Mb) or pools of CTCs (~10/Mb). Lohr et al. further observe that the estimated FPR decreases as a function of pooling, however intersecting SNVs across independent libraries was ultimately necessary for accurate SNV calls in their study (Supplementary Figs. 9, 11a in Lohr et al.). Although the FPR estimated by Lohr et al. is an upper bound that does not account for real divergence from the primary tumor, and Faugeroux et al. use Ampli1 WGA rather than MDA, the burden of proof is on the authors to establish that their highly inflated estimates are real. Without such evidence, the authors should either employ a similar approach to the one established by Lohr et al. or explain to potential readers why their current results are believed to be correct while also describing the serious caveats noted above.
2. As further evidence that the authors' CTC mutation estimates are inflated, previous studies of metastatic prostate cancer have identified considerably higher concordance between CTC and primary tumor genotypes, whether considering SNVs (51 to 70% concordant with primaries; Lohr et al. 2014) or copy number variants (91%, doi: 10.1093/annonc/mdz248). In stark contrast, 97.7% of SNVs called in CTCs in this study are private to a single CTC sample. The authors favor a biological interpretation of this striking result rather than the possibility of a technical artifact. This should be addressed.

Minor comments:

1. Replace commas with decimals when representing non-integer numbers (e.g., Fig. S5A,C).
2. Fig. S6: The authors need to describe the underlying data and processing for this figure.

Manuscript: NCOMMS-19-01047A

Title: "Genetic characterization of a Unique Neuroendocrine Transdifferentiation Prostate Circulating Tumor Cell - Derived eXplant (CDX) Model".

Response to Reviewers

Reviewer #2

Overall, the authors have improved the manuscript and from a technical standpoint I believe they have addressed my concerns. However, the main concern I have is that this is an N=1 report and generalizability about the insight of genomic evolution remains to be determined. Also, I find there is still a sense of overly ambitious claims. For example, the statement "Here, we report for the first time the feasibility of using DLA-increased CTC yield for CDX establishment." I don't think one can say a single CDX establishment can be sufficient to demonstrate "increased CTC yield for CDX establishment"

We fully understand this comment and have toned down the generalization of our findings. We recognize that the genomic evolution of CRPC into CRPC-NE described in our study cannot be generalized based on our only model.

We have softened our conclusions in the discussion section page 35 lines 12-13 ("we can determine the order of acquisition..."has been adjusted to "This in-depth genomic analysis suggests an order of acquisition...") and page 39 lines 8 and 9 ("the finding that *TP53* loss precedes loss of *PTEN* and *RB1* uncovers a possible sequence" has been adjusted to "the finding that *TP53* loss precedes loss of *PTEN* and *RB1* suggests a possible sequence...").

Also, we fully recognize that a single CDX establishment is not sufficient to demonstrate "DLA-increased CTC yield for CDX establishment". We have now replaced "we report for the first time the feasibility of using DLA-increased CTC yield for CDX establishment" with "we report for the first time the use of DLA for CDX establishment, a result which suggests that DLA-increased CTC yield may enhance the chances of CDX success" (discussion section page 36 lines 5-7).

Reviewer #3

This reviewer is satisfied with all responses by the authors. The manuscript is much improved.

We thank the reviewer for his comment.

Figure 2A could be improved for presentation. The gene names are too small to read as is. Perhaps they could either use the current 2A in the supplement and create a condensed version where only the labels of categories are presented. Regardless, as this is an important figure some additional thought should be given to presenting figure 2.

We have now provided a simplified version of Figure 2A where only the pathways and the number of genes analyzed in each pathway are mentioned. We have also included an additional table that presents the list of the 250 functional genes that are relevant for CRPC-NE progression and significantly deregulated in the CDX and CDX-derived cell line compared to LNCaP cells. The Log2 fold change, the pvalue and qvalue are presented for each gene (Supplementary Table 3).

Reviewer #4

In their revised manuscript, Faugeroux and colleagues have addressed serious issues with their tumor genotyping criteria, resulting in more credible results. Although the manuscript has improved, the substantial discrepancy between the large number of somatic variants per CTC (median of 338 SNVs per exome) and the more realistic number of somatic variants per primary tumor (median of 28 SNVs per exome) suggests that more rigorous criteria are needed for genotyping the latter, as detailed below.

Major comments:

1. In their rebuttal, the authors cite Beltran et al. Nat Med 2016 as a reference for the median number of exonic SNVs expected in metastatic tumor samples from patients with castration-resistant neuroendocrine prostate cancer (n = 41). It is therefore surprising that the authors are seemingly content with a nearly 10-fold higher number of SNVs in their CTC exomes. In fact, in a highly relevant prior study that is not cited by the authors (PMID 24752078), Lohr and colleagues show that WGA using MDA significantly inflates the false positive rate of calling SNVs from CTCs, whether as single CTCs (~25/Mb) or pools of CTCs (~10/Mb). Lohr

et al. further observe that the estimated FPR decreases as a function of pooling, however intersecting SNVs across independent libraries was ultimately necessary for accurate SNV calls in their study (Supplementary Figs. 9, 11a in Lohr et al.). Although the FPR estimated by Lohr et al. is an upper bound that does not account for real divergence from the primary tumor, and Faugeroux et al. use Ampli1 WGA rather than MDA, the burden of proof is on the authors to establish that their highly inflated estimates are real. Without such evidence, the authors should either employ a similar approach to the one established by Lohr et al. or explain to potential readers why their current results are believed to be correct while also describing the serious caveats noted above.

We fully understand this constructive comment and have modified the analysis of CTCs accordingly. To estimate the false positive rates in CTC samples, we have now adopted a conservative approach similar to that reported by Lohr *et al.* and considered as false positive all events not found in the primary tumor and the CDX (Patients and Methods section page 15 lines 9-12). The choice to include events not found in the CDX is justified by the high VAF found in the CDX. As now mentioned in the Results section page 28 lines 19-23 and page 29 lines 1-2 and shown in Supplementary Figure 5B, our pools of five CTCs exhibited FPR values ranging from 7 per Mb to 21 per Mb, which supports results reported by Lohr *et al.* Therefore, we have now applied additional criteria to generate a high-confidence set of variants from CTCs. We adopted an approach inspired from the one used by Lohr *et al.* and Chemi *et al.* (where Ampli1 WGA was used, Nat Med 2019). High-confidence CTC variants had to be present in either the primary tissue (at least 1 primary tumor specimen) or the CDX or at least one other CTC sample (Patients and Methods section page 14 lines 9-11, Results section page 30 lines 8-17, Supplementary Table 5). The number of high-confidence variants present in CTCs has been therefore considerably decreased with a set of 62 high-confidence somatic variants identified in the six CTC samples.

The reference Lohr *et al.* has been included in the discussion section page 37 line 19.

Lohr JG, Adalsteinsson VA, Cibulskis K, Choudhury AD, Rosenberg M, Cruz-Gordillo P, et al. Whole-exome sequencing of circulating tumor cells provides a window into metastatic prostate cancer. Nat Biotechnol. 2014;32(5):479-84.

Chemi F, Rothwell DG, McGranahan N, Gulati S, Abbosh C, Pearce SP, et al. Pulmonary venous circulating tumor cell dissemination before tumor resection and disease relapse. Nat Med. 2019;25(10):1534-9.

2. As further evidence that the authors' CTC mutation estimates are inflated, previous studies of metastatic prostate cancer have identified considerably higher concordance between CTC and primary tumor genotypes, whether considering SNVs (51 to 70% concordant with primaries; Lohr et al. 2014) or copy number variants (91%, doi: 10.1093/annonc/mdz248). In

stark contrast, 97.7% of SNVs called in CTCs in this study are private to a single CTC sample. The authors favor a biological interpretation of this striking result rather than the possibility of a technical artifact. This should be addressed.

Taking into account the new criteria mentioned above, we now find 40% concordance between CTCs and primary tumor mutations, which supports results reported by Lohr *et al.* This is now presented in the Results section page 30 lines 8-17 and in the Discussion section page 37 lines 16-23 and page 38 lines 1-2.

Minor comments:

1. Replace commas with decimals when representing non-integer numbers (e.g., Fig. S5A,C).

Commas have been removed and numbers rounded up (Supplementary Tables 4 and 5).

2. Fig. S6: The authors need to describe the underlying data and processing for this figure. We have now described the underlying data and processing for Supplementary Figure 6.

“To identify samples with similar mutational profiles, we selected all variants present in at least two samples, and classified the samples based on their VAF across these mutations using principal component analysis and hierarchical clustering (Ward method, cosine distance). This method allows to regroup samples sharing the same mutational profile”.

The corresponding Patients and Methods section has been modified accordingly (page 16 lines 22-23 and page 17 lines 1-3).

REVIEWERS' COMMENTS:

Reviewer #4 (Remarks to the Author):

The authors have satisfactorily addressed my remaining concerns.